# Large-scale cross-ancestry genome-wide meta-analysis of serum urate

Chamlee Cho [1,13], Beomsu Kim [1,13], Dan Say Kim[1,13], Mi Yeong Hwang [2], Injeong Shim [1], Minku Song[1], Yeong Chan Lee [3], Sang-Hyuk Jung [4], Sung Kweon Cho [5], Woong-Yang Park[6], Woojae Myung [7], Bong-Jo Kim[2], Ron Do [8,9], Hyon K. Choi[10], Tony R. Merriman [11,12], Young Jin Kim [2,14] ✉ & Hong-Hee Won [1,6,14] ✉

Hyperuricemia is an essential causal risk factor for gout and is associated with cardiometabolic diseases. Given the limited contribution of East Asian ancestry to genome-wide association studies of serum urate, the genetic architecture of serum urate requires exploration. A large-scale cross-ancestry genome-wide association meta-analysis of 1,029,323 individuals and ancestry-specific meta-analysis identifies a total of 351 loci, including 17 previously unreported loci. The genetic architecture of serum urate control is similar between European and East Asian populations. A transcriptome-wide association study, enrichment analysis, and colocalization analysis in relevant tissues identify candidate serum urate-associated genes, including *CTBP1*, *SKIV2L*, and *WWP2*. A phenome-wide association study using polygenic risk scores identifies serum urate-correlated diseases including heart failure and hypertension. Mendelian randomization and mediation analyses show that serum urate-associated genes might have a causal relationship with serum urate-correlated diseases via mediation effects. This study elucidates our understanding of the genetic architecture of serum urate control.

Serum urate (SU) is known to cause gout if a high SU level (hyperuricemia) is maintained[1]. It is associated with several diseases, including nephrolithiasis, hypertension, and cardiovascular disease[2–4]. According to guidelines from the American College of Rheumatology[5], urate-lowering therapeutics (ULTs) are strongly recommended to decrease the risk of gout flares after gout diagnosis. Despite the importance of SU in managing related diseases, only five Food and Drug Administration (FDA) approved and manufactured ULTs are currently available: allopurinol, febuxostat, probenecid, rasburicase, and pegloticase. These medications have multiple limitations, including severe allergic reactions, increased risk of cardiovascular death, drug-drug interactions, and high costs[6]; therefore, novel ULTs are required. Given the

[1]Department of Digital Health, Samsung Advanced Institute for Health Sciences and Technology (SAIHST), Sungkyunkwan University, Samsung Medical Center, Seoul, Republic of Korea. [2]Division of Genome Science, Department of Precision Medicine, National Institute of Health, Cheongju-si, Chungcheongbuk-do, Republic of Korea. [3]Research Institute for Future Medicine, Samsung Medical Center, Seoul, Republic of Korea. [4]Department of Biostatistics, Epidemiology and Informatics, Perelman School of Medicine, University of Pennsylvania, Philadelphia, PA, USA. [5]Department of Pharmacology, Ajou University School of Medicine (AUSOM), Suwon, Republic of Korea. [6]Samsung Genome Institute, Samsung Medical Center, Sungkyunkwan University School of Medicine, Seoul, Republic of Korea. [7]Department of Neuropsychiatry, Seoul National University Bundang Hospital, Seongnam, Republic of Korea. [8]The Charles Bronfman Institute for Personalized Medicine, Icahn School of Medicine at Mount Sinai, New York, NY, USA. [9]Department of Genetics and Genomic Sciences, Icahn School of Medicine at Mount Sinai, New York, NY, USA. [10]Division of Rheumatology, Allergy and Immunology, Massachusetts General Hospital, Harvard Medical School, Boston, MA, USA. [11]Biochemistry Department, University of Otago, Dunedin, New Zealand. [12]Division of Clinical Immunology and Rheumatology, University of Alabama at Birmingham, Birmingham, AL, USA. [13]These authors contributed equally: Chamlee Cho, Beomsu Kim, Dan Say Kim. [14]These authors jointly supervised this work: Young Jin Kim and Hong-Hee Won. ✉e-mail: inthistime@korea.kr; wonhh@skku.edu

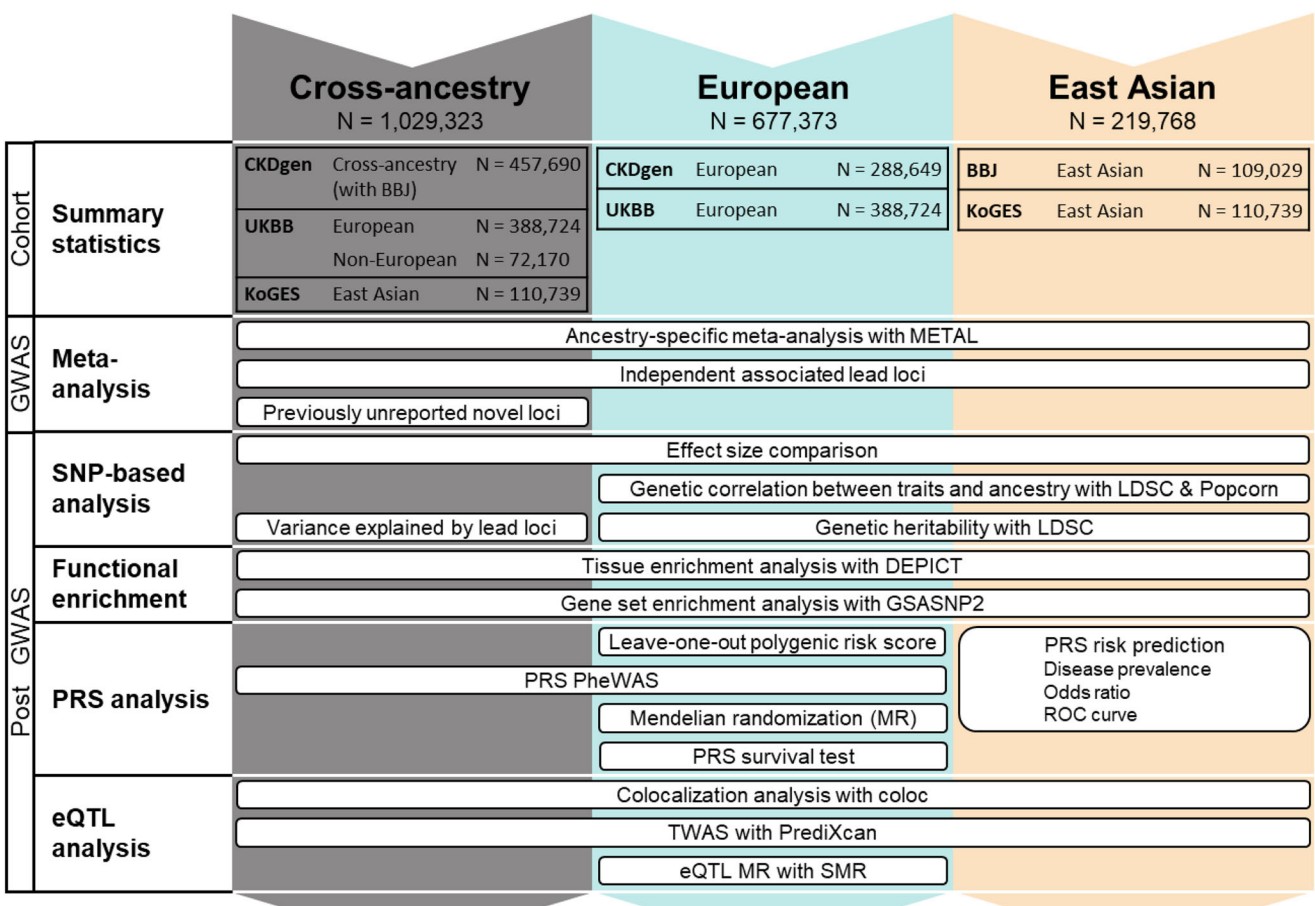

**Fig. 1 | Study overview.** Overview of this study. Six GWAS summary statistics were used for the meta-analysis of the ancestry-specific study. In the cross-ancestry study, previously unreported loci were identified by a meta-analysis of CKDgen, UKBB, and KoGES summary statistics. The European ancestry study performed a meta-analysis of CKDgen (European) and UKBB (European) summary statistics, whereas the East Asian ancestry study performed a meta-analysis of BBJ and KoGES. In this ancestry-specific meta-analysis, post-GWAS, such as functional enrichment, PRS, and eQTL analyses, were performed separately for each meta-analysis. GWAS genome-wide association study, SNP single-nucleotide polymorphism, LDSC linkage disequilibrium score regression, PRS polygenic risk score, pheWAS phenome-wide association study, MR Mendelian randomization, SMR summary-based MR, UKBB UK BioBank, CKDgen Chronic Kidney Disease Genetics Consortium, KoGES Korean Genome and Epidemiology study, BBJ Biobank Japan.

heritable nature of SU (30–70%)[7], revealing the underlying genetics of SU should enhance the understanding of SU biology and the pathogenesis of related diseases.

Genes related to SU, including *SLC2A9* (*GLUT9*), *ABCG2*, and *SLC22A12* (*URAT1*), have been discovered in several genome-wide association studies (GWAS)[8–12]. More SU-associated variants and genes have been discovered through a large-scale cross-ancestry meta-analysis, and the causality and pleiotropy between SU and several cardiometabolic traits were evaluated. Additionally, a novel missense mutation in *HNF4A* (p.Thr139Ile) involved in urate homeostasis had its function experimentally validated[13]. However, the results were derived from data containing a disproportionately large number of individuals of European ancestries. Inequity in disease risk prediction for non-European populations results from this Eurocentric bias, implicating the need for additional genomic studies conducted on non-European populations[14].

In this study, we aimed to identify novel variants, genes, and pathways associated with SU through a large-scale cross-ancestry meta-analysis of 1,029,323 individuals of multiple ancestries (Europeans = 677,373, East Asians = 219,768, others = 132,182), followed by a functional assessment of the results comprising colocalization, transcriptome-wide association study (TWAS), and functional enrichment analyses. A phenome-wide association study (PheWAS) using the polygenic risk score (PRS) was performed to understand the

genetic relationship of various traits with SU. Potential causal relationships between SU, heart failure, and hypertension were examined using Mendelian randomization analysis. To identify potential therapeutic targets of ULTs for the above three diseases, summary-based Mendelian randomization (SMR) and mediation analyses were performed.

## Results

### Cross-ancestry and ancestry-specific GWAS for SU

We performed three genome-wide meta-analyses (cross-ancestry, European, and East Asian) for SU using GWAS summary statistics from the Chronic Kidney Disease Genetics Consortium (CKDGen, $N = 457,690$)[13], UK Biobank (UKBB, $N_{EUR} = 388,724$, $N_{non-EUR} = 72,170$)[15], Biobank Japan (BBJ, $N = 109,029$)[16], and the Korean Genome and Epidemiology Study (KoGES, $N = 110,739$) genotyped by the Korea Biobank Array (KBA) project[17,18] (Figs. 1 and 2, Supplementary Fig. 1, and Supplementary Data 1). To ascertain the robustness of the PC values provided by the UKBB in accounting for population stratification, we also performed GWAS using PC values derived from European and non-European populations, respectively (Methods). GWAS results adjusted for the newly calculated PCs were highly consistent with the original GWAS results adjusted for the provided PCs (Supplementary Figs. 2 and 3). We identified 351, 269, and 90 lead signals in the cross-ancestry

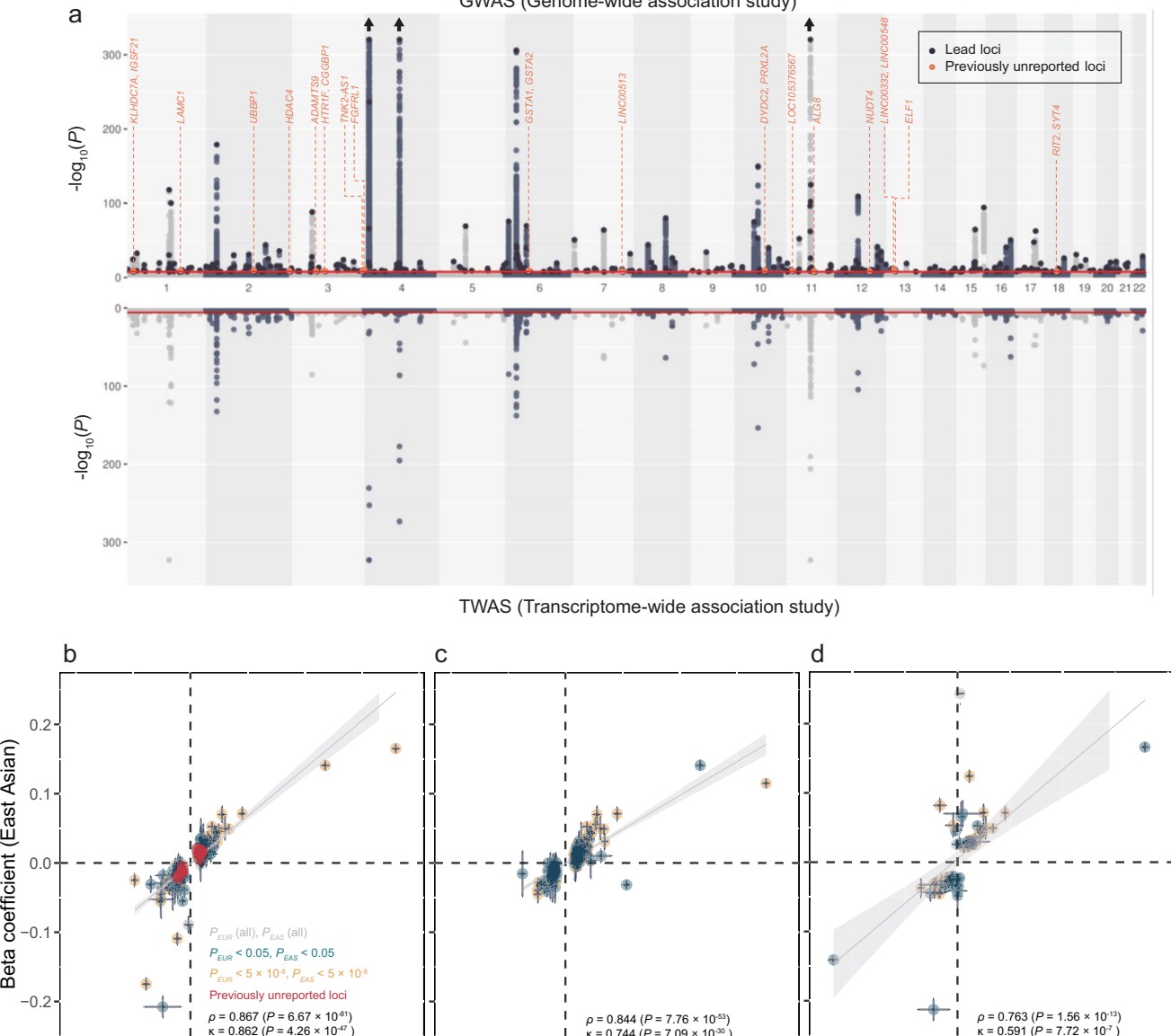

**Fig. 2 | Mirrored Manhattan plots of GWAS and TWAS in the cross-ancestry meta-analysis and comparison of the variant effect sizes for each ancestry-specific meta-analysis. a** GWAS and TWAS mirrored Manhattan plots for cross-ancestry study. The upper plot represents the GWAS result, and lower plot represents the TWAS result. The red line in the upper graph represents the GWAS significance cutoff ($P = 5 \times 10^{-8}$), and that in the lower graph represents the TWAS significance cutoff after Bonferroni's correction ($P = 2.31 \times 10^{-6}$). The orange dots represent previously unreported loci and the genes mapped or associated with those loci are labeled. TWAS associations for all 49 tissues are shown.

**b–d** Comparison of effect sizes of the lead variants between the European and East Asian ancestries. **b** Cross-ancestry meta-analysis lead variants. **c** European ancestry meta-analysis lead variants. **d** East Asian ancestry meta-analysis lead variants. Each point represents the beta coefficient of the lead variant. The horizontal lines in the points reflect its standard deviation in the European meta-analysis, and the vertical lines represent the standard deviation in the East Asian meta-analysis. *P*-values were determined using a two-sided test. GWAS, genome-wide association study; TWAS, transcriptome-wide association study.

($N = 1,029,323$), European ancestry ($N = 677,373$), and East Asian ancestry ($N = 219,768$) analyses, respectively (Supplementary Data 2, 3, and 4). The genetic correlation of SU between the two ancestries was estimated to be high ($\rho_{ge} = 0.942$, standard error [s.e.] = 0.079) using Popcorn[19] (Supplementary Data 5). The effect size and direction of lead variants from each of the ancestry meta-analysis results showed modest to high correlations between European and East Asian populations ($\rho = 0.763$–$0.867$, $\kappa = 0.591$–$0.862$, Fig. 2b–d). Of the significant lead loci, 58, seven, and one were identified only in the cross-ancestry, European-specific, and East Asian-specific analyses, respectively (Supplementary

Data 6). We compared the effect size and the direction of the lead variants in the cross-ancestry meta-analysis with those in each of the four genetically distinct groups (India, Italy, Nigeria, and Poland) and found significant positive correlations ($\rho = 0.426$–$0.77$, $\kappa = 0.376$–$0.626$, Supplementary Fig. 4).

The cross-ancestry meta-analysis additionally identified 17 loci that were previously unreported in the GWAS Catalog and SU GWASs (Table 1 and Supplementary Fig. 5). Of these, six loci were previously associated with SU-associated traits, such as triglyceride, chronic obstructive pulmonary disease (COPD), and type 2 diabetes (T2D)[20–22].

**Table 1 | Summary of 17 previously unreported loci**

| SNP info | | | | GWAS catalog | | | GTEx | |
|---|---|---|---|---|---|---|---|---|
| SNP | Position | Effect | P value | Consequence | Mapped gene | Reported trait | Gene | Tissue |
| rs2992756[b] | 1:18807339 | 0.0132 | $1.15 \times 10^{-8}$ | Intergenic | KLHDC7A IGSF21 | Breast cancer eGFR | KLHDC7A | Thyroid Liver |
| rs4129858[b] | 1:183004334 | −0.0131 | $2.73 \times 10^{-9}$ | Intron | LAMC1 | Blood protein levels Type 2 diabetes | LAMC1 LAMC1-AS1 | Artery (Tibial) Adipose (Subcutaneous) |
| rs6708702[b] | 2:137074132 | 0.015 | $3.19 \times 10^{-9}$ | Intergenic | UBBP1 | eGFR BUN | THSD7B | Artery (Aorta) |
| rs12623489[a] | 2:240222564 | −0.0195 | $1.12 \times 10^{-8}$ | Intron | HDAC4 | Triglycerides HDL eGFR SBP DBP | | |
| rs13096863[a] | 3:64651920 | −0.0147 | $6.51 \times 10^{-9}$ | Intron | ADAMTS9 | Waist-hip ratio ABSI Type 2 diabetes Hip circumference SBP DBP CAD BFP BUN Pulse pressure | ADAMTS9 | Cells (Cultured fibroblasts) |
| rs13100870 | 3:88099788 | −0.0196 | $9.43 \times 10^{-9}$ | | | BMI | C3orf38 | Artery (Tibial) Adipose (Subcutaneous) |
| rs34221697[a] | 3:195635432 | 0.0463 | $3.20 \times 10^{-9}$ | nc-transcript variant 2KB upstream variant | TNK2-AS1 TNK2 | BUN eGFR SCL | TNK2 | Artery (Tibial) Adipose (Subcutaneous) |
| rs4647939[a] | 4:1019312 | 0.015 | $8.85 \times 10^{-11}$ | 3'-UTR | FGFRL1 | Type 2 diabetes eGFR | UVSSA FGFRL1 CRIPAK | Artery (Aorta) Artery (Tibial) Thyroid |
| rs9395816 | 6:52637594 | −0.0154 | $5.67 \times 10^{-9}$ | | | Triglyceride Serum urea level | GSTA1 GSTA2 | Adrenal Gland Liver |
| rs125124[b] | 7:130584684 | 0.0127 | $4.00 \times 10^{-8}$ | Intron | LINC00513 LINC-PINT | BCC SAL STP | LINC-PINT | Cells (Cultured fibroblasts) |
| rs17616958[a] | 10:82137461 | −0.0234 | $3.60 \times 10^{-9}$ | | | COPD | MBL1P PLAC9 | Artery (Tibial) Adipose (Subcutaneous) |
| rs34869311 | 11:15769438 | 0.015 | $3.19 \times 10^{-9}$ | Intron | LOC105376567 | | | |
| rs529343[a] | 11:77843719 | 0.0159 | $2.82 \times 10^{-8}$ | Intron | ALG8 | DBP BMI | NDUFC2-KCTD21 | Heart (Atrial Appendage) Adipose (Subcutaneous) Brain (Cortex) |
| rs12422756[a] | 12:93789526 | 0.0185 | $1.99 \times 10^{-9}$ | Intron | NUDT4 | Blood cell related | UBE2N NUDT4 | Cells (Cultured fibroblasts) |
| rs112533663 | 13:40767072 | 0.0199 | $2.20 \times 10^{-11}$ | Intron | LOC124903162 | | LINC00332 FOXO1 | Pancreas Cells (Cultured fibroblasts) |
| rs73176948[b] | 13:41611271 | −0.0286 | $7.90 \times 10^{-9}$ | Intron | ELF1 | AST ALT | WBP4 | Heart (Left Ventricle) Adipose (Subcutaneous) Thyroid |
| rs1518149[b] | 18:40732054 | −0.0123 | 2.35E−08 | | | | | |

SNP single-nucleotide polymorphism, GWAS genome-wide association study, eGFR estimated glomerular filtration rate, HDL high-density lipoprotein cholesterol, SBP systolic blood pressure, DBP diastolic blood pressure, ABSI a body shape index, BFP body fat percentage, CAD coronary artery disease, BUN blood urea nitrogen levels, SCL serum creatinine levels, BCC basal cell carcinoma, SAL serum albumin levels, STP serum total protein level, COPD chronic obstructive pulmonary disease, BMI body mass index, CRP C-reactive protein, AST aspartate aminotransferase, ALT alanine aminotransferase, eQTL expression quantitative trait loci.
[a]Mapped genes and eQTL genes are associated with reported traits.
[b]Lead SNP of loci is associated with reported traits.

In addition, the nearest genes or expression quantitative loci (eQTL) genes in seven loci were previously associated with SU-associated traits, such as *ADAMTS9* associated with coronary artery disease and *MBL1P* associated with COPD[23,24]. The remaining four loci were previously unreported.

**Heritability estimation and genetic correlation**
We estimated single-nucleotide polymorphism (SNP)-based heritability using linkage disequilibrium score regression (LDSC) v1.0.1[25] for European and East Asian populations (Methods and Supplementary Data 7). The SNP-based heritability values for the European and East Asian cohorts were 8.74 and 11.87%, respectively. The proportion of SU variance explained by the lead SNPs in our cross-ancestry GWAS was 8.36% compared to 7.7% in a previous cross-ancestry GWAS[13].

We estimated the genetic correlation between SU and other traits using LDSC v1.0.1 (Supplementary Fig. 6). In the European GWAS, 63 of 320 traits showed significant genetic correlations that passed the false discovery rate (FDR) threshold ($P < 0.0071$). The most significant

positive correlation was renal failure ($r_g = 0.50$, $P = 1.36 \times 10^{-5}$) in the physical health category, followed by T2D ($r_g = 0.36$, $P = 7.48 \times 10^{-25}$) and hypertension ($r_g = 0.31$, $P = 1.17 \times 10^{-21}$). In the laboratory and physical findings category, high-density lipoprotein (HDL) cholesterol (UKBB, $r_g = -0.3$, $P = 4.11 \times 10^{-25}$) and testosterone ($r_g = -0.22$, $P = 3.41 \times 10^{-12}$) were negatively correlated with SU. In the East Asian GWAS, 26 of 181 traits showed significant genetic correlations that passed the FDR threshold ($P < 0.0083$). The most significant positive correlation was C-reactive protein ($r_g = 0.3$, $P = 1.3 \times 10^{-3}$) in the

laboratory and physical findings category, followed by triglycerides (AGEN, $r_g = 0.27$, $P = 3.0 \times 10^{-4}$) and y-glutamyl transferase ($r_g = 0.24$, $P = 3.0 \times 10^{-4}$). In the physical health category, estimated glomerular filtration rate ($r_g = -0.22$, $P = 1.39 \times 10^{-5}$) and HDL cholesterol ($r_g = -0.15$, $P = 7.6 \times 10^{-3}$) were negatively correlated with SU.

### Tissue and gene set enrichment of SU GWAS

We investigated the tissues in which the genes of SU-associated loci were enriched according to the physiological system category (Fig. 3a,

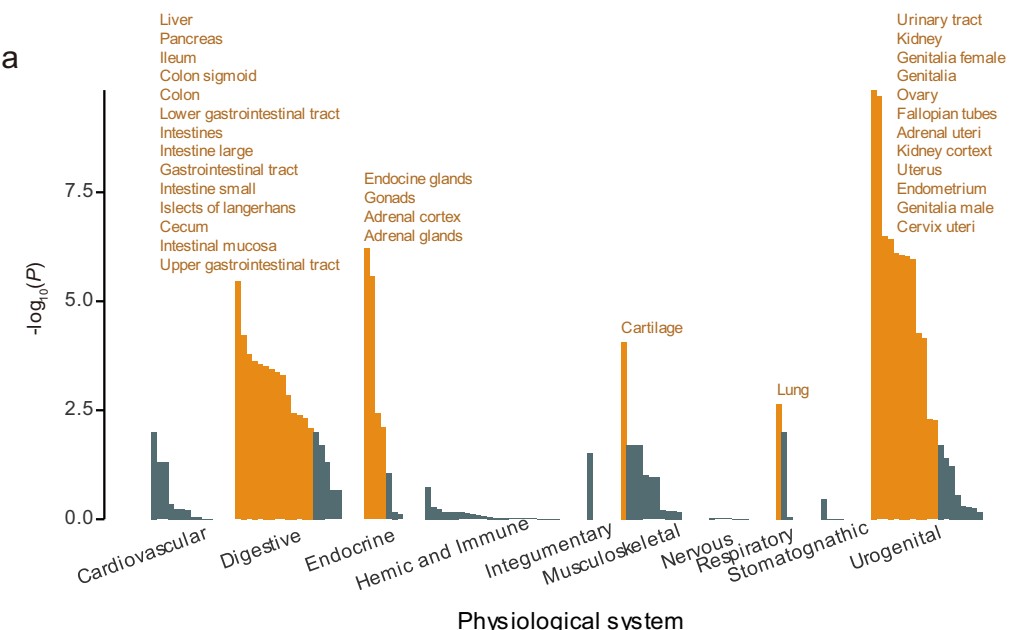

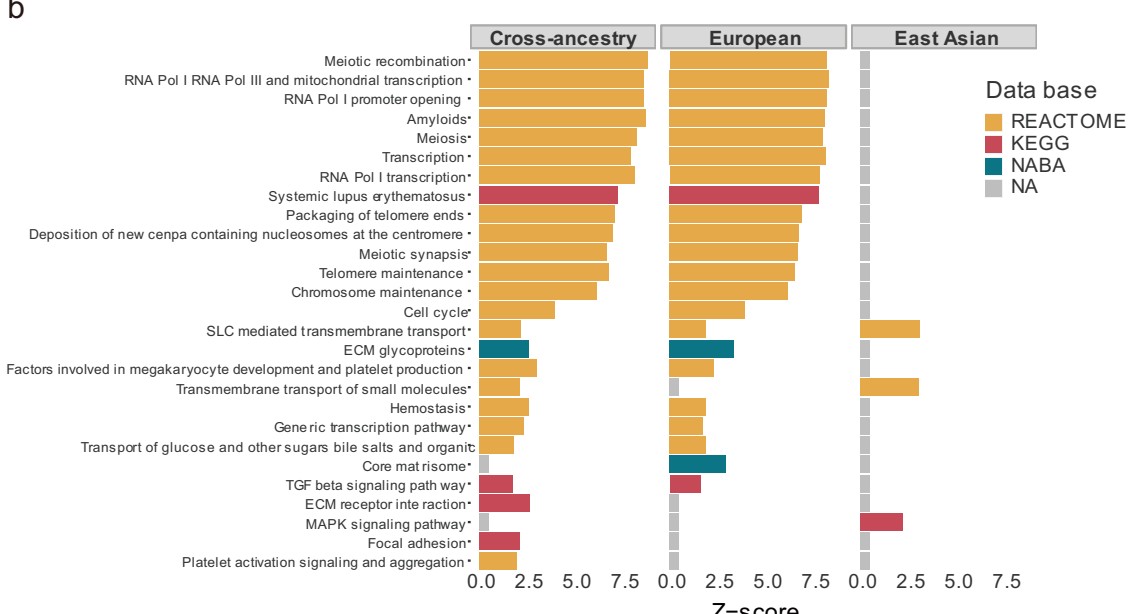

**Fig. 3 | The tissue enrichment analysis in cross-ancestry meta-analysis and gene-set enrichment analysis in ancestry-specific meta-analysis for serum urate-associated loci. a** Tissue enrichment related to SU-associated loci in cross-ancestry meta-analysis is shown by tissue groups into physiological systems. The *x*-axis represents tissues grouped by physiological systems, and the *y*-axis represents -log$_{10}$ (*P*-value). The orange color indicates significantly enriched tissues and labels (FDR < 0.05). *P*-values were determined using a two-sided test. **b** For the meta-

analysis results for each ancestry, gene-set enrichment was performed using GSA-SNP2. The enrichment in the canonical pathway gene sets of databases such as KEGG (Kyoto Encyclopedia of Genes and Genomes), NABA (Matrisome Project), and REACTOME (Reactome Project) was investigated using MSigDb c2.cp.v6.2. Only significantly enriched gene sets with *q*-value < 0.25 are shown after FDR correction. SU serum urate, FDR false discovery rate.

Supplementary Fig. 7, and Supplementary Data 8). A total of 40, 36, and 29 tissues showed significant enrichment in cross-ancestry, European, and East Asian, respectively. Overall, various tissues of the urogenital, digestive, and endocrine systems were significantly enriched. The urinary tract of the urogenital system showed the strongest enrichment ($P_{CROSS} = 1.43 \times 10^{-10}$, $P_{EUR} = 3.99 \times 10^{-9}$, $P_{EAS} = 2.54 \times 10^{-4}$). Cardiovascular system-related tissues, such as heart valves, were significantly enriched in the European ancestry only ($P = 6.07 \times 10^{-4}$). The fetal blood of the hematologic and immune system ($P = 9.31 \times 10^{-3}$) and the nasal mucosa of the respiratory system ($P = 3.65 \times 10^{-3}$) were significantly enriched in the East Asian ancestry only.

We conducted gene set enrichment analysis using genes in the loci significantly associated with SU. Significant gene sets that passed the FDR correction (FDR ≤ 0.25) were selected from the results obtained using GSA-SNP2 (released 2020-09-01) (Fig. 3b). As in the tissue enrichment analysis, more gene set enrichment results were identified in the cross-ancestry GWAS (Supplementary Data 9). We identified additional results of the Kyoto Encyclopedia of Genes and Genomes (KEGG) pathway related to high-level functions of biological systems. For example, the "systemic lupus erythematosus" gene set in the KEGG pathways was significantly enriched, and the relationship between this pathway and SU has been reported in previous clinical studies[26,27].

## Colocalization with eQTL in glomerular and tubulointerstitial tissues

To further understand the functional roles of the identified loci and identify candidate causal genes, specifically expressed genes in glomerular (GLOM) and tubulointerstitial (TUBE) tissues were analyzed in terms of colocalization with the identified loci in this study. In total, 173 genes in GLOM and TUBE tissues from the Nephrotic Syndrome Rare Disease Clinical Research Network III (NEPTUNE)[28] were colocalized with our GWAS results (posterior probability for colocalization [PP.H4] >0.8); 54 and 148 colocalized genes in GLOM and TUBE tissues, respectively (Fig. 4a and Supplementary Data 10, 11, and 12). These genes were colocalized with 159, 110, and 48 GWAS SU association signals in the cross-ancestry, European, and East Asian cohorts, respectively. Of the colocalized genes, 27 and 9 were identified in the European and East Asian ancestry only, respectively. For most of the genes colocalized in both cross-ancestry and European results, the number of variants included in the credible set was less in the cross-ancestry than in the European set; 28 of 30 and 40 of 50 genes in the GLOM and TUBE tissues showed identical or reduced size of the 95% credible set, respectively (Fig. 4b). By conducting a cross-ancestry meta-analysis, 68 more colocalized genes were identified than in the European meta-analysis, and 52 of those genes were cross-ancestry-specific. Cross-ancestry revealed colocalization signals that were not found in ancestry-specific analyses; although rs28362590, a lead variant near *MXD3* in the cross-ancestry ($\beta = 0.021$, SE = 0.0025, $P = 4.15 \times 10^{-17}$), showed genome-wide significance in the European ($\beta = 0.019$, SE = 0.0031, $P = 1.80 \times 10^{-9}$) and East Asian ($\beta = 0.023$, SE = 0.0036, $P = 1.93 \times 10^{-10}$) cohorts, *cis*-eQTLs of *MXD3* in the TUBE tissue were colocalized only with SU GWAS results in the cross-ancestry (PP.H4 = 0.930), but not in the European (PP.H4 = 0.076) and East Asian (PP.H4 = 0.770) cohorts (Fig. 4c).

## TWAS

We conducted a TWAS using GWAS results for each ancestry-specific meta-analysis result to identify genes whose predicted gene expression levels were associated with SU (Methods, Fig. 2a, and Supplementary Fig. 1). TWAS was conducted to determine the association between SU-associated loci and Genotype-Tissue Expression (GTEx) v8 eQTL results in 49 tissues[29]. A total of 1,111 genes were significantly associated with SU across all the tissues (Supplementary Data 13, 14, and 15). While 178 genes were commonly significant in the three meta-

analyses, 183 of 945, 83 of 780, and 75 of 334 significant genes were only significant in the cross-ancestry, European, and East Asian populations, respectively.

## PheWAS and survival analysis using PRS

We calculated the SU PRS for UKBB individuals using each meta-analysis from the cross-ancestry and European cohorts as reference summary statistics (Methods and Supplementary Fig. 8a). PheWAS was conducted across 1621 UKBB phecodes using cross-ancestry and European ancestry SU PRS. A total of 129 and 142 phenotypes were significant in PheWAS using cross-ancestry and European PRSs, respectively (Fig. 5 and Supplementary Fig. 9). Among these, 133 phenotypes, including gout and heart failure, were commonly associated. Three were significant only when the cross-ancestry PRS was applied, including pulmonary heart disease, erythematous conditions, other alveolar and parietoalveolar pneumonopathies, and 16 were significant only when the European PRS was applied (Supplementary Data 16 and 17). In addition, we conducted the PheWAS with the cross-ancestry SU PRS and East Asian SU PRS on the Korean participants to investigate the similarities and differences with the European results (Methods and Supplementary Fig. 8b). Among the 37 self-reported diseases, gout and hypertension were significantly associated with the SU PRS (Supplementary Data 18).

To investigate the association between the polygenic risk for SU and the risk of gout, heart failure, and hypertension, we performed survival analyses using cross-ancestry and the European PRS on 380,213 participants with gout, 331,432 participants with heart failure, and 357,453 participants with hypertension who did not take ULT-related medications at enrollment. Compared to the group with low PRS, those with high PRS presented a higher absolute incidence rate for the three traits in both the cross-ancestry and European PRS analyses (Supplementary Data 19 and 20). During the median follow-up period of 12.80, 12.76, and 11.62 years for gout, heart failure, and hypertension, respectively, we evaluated the association of SU PRS with the traits using Cox proportional hazard regression models. Cross-ancestry and European ancestry SU PRS were significantly associated with gout risk (cross-ancestry, hazard ratio [HR] = 1.63; 95% confidence interval [95% CI] = 1.592–1.67; $P < 2.00 \times 10^{-16}$; European, HR = 1.641; 95% CI = 1.603–1.681; $P < 2.00 \times 10^{-16}$). Both SU PRS were also significantly associated with risk of heart failure (cross-ancestry, HR = 1.056; 95% CI = 1.036–1.078; $P < 2.00 \times 10^{-16}$; European, HR = 1.061; 95% CI = 1.04–1.082; $P < 2.00 \times 10^{-16}$) and hypertension (cross-ancestry, HR = 1.065; 95% CI = 1.056–1.073; $P < 2.00 \times 10^{-16}$; European, HR = 1.065; 95% CI = 1.057–1.074; $P < 2.00 \times 10^{-16}$). Compared to the group with lower SU PRS, the HR of incident gout, heart failure, and hypertension was higher in the higher PRS group in both cross-ancestry and European ancestry SU PRSs. For example, participants in the very high (99th percentile) SU cross-ancestry PRS group showed 7.00-, 1.37-, and 1.34-times higher risk of gout, heart failure, and hypertension, respectively, than participants in the low (0–19th percentile) SU cross-ancestry PRS group. These results were similar to those of the analysis of participants of European ancestry (Supplementary Data 21). The Kaplan–Meier survival curve showed a similar result (Supplementary Fig. 10).

## Disease risk prediction in an independent Korean population using the cross-ancestry and East Asian SU PRS

SU PRSs of KoGES individuals were calculated using cross-ancestry and East Asian meta-analyses. These two SU PRSs were used to predict disease risk in an independently genotyped Korean sample set (KoGES). The prevalence of hypertension and gout increased according to the SU PRS groups, and these differences were greater when the cross-ancestry meta-analysis was used to calculate the SU PRS. When cross-ancestry PRS was applied, the prevalence of hypertension was 21.9, 20.8, and 18.2% in the high, intermediate, and low groups,

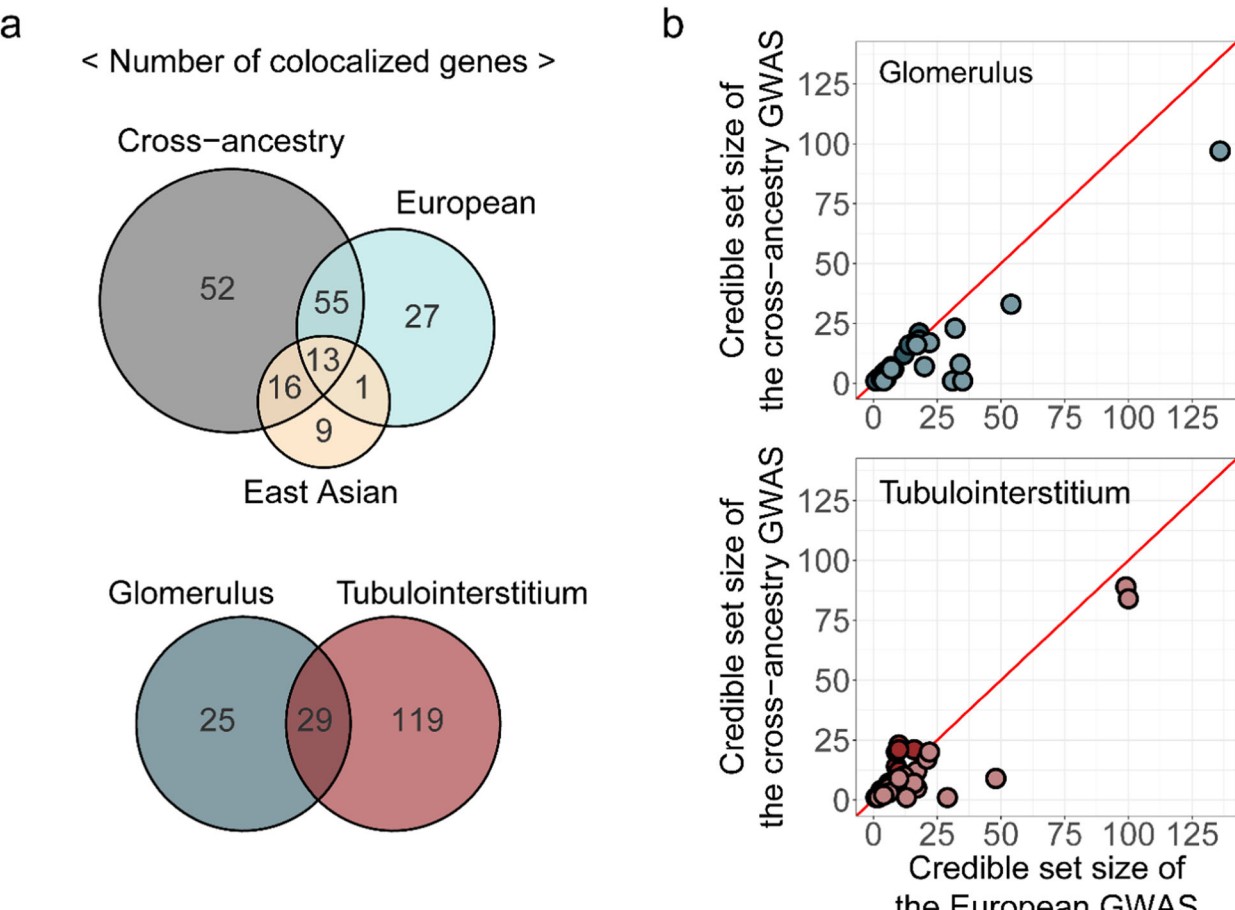

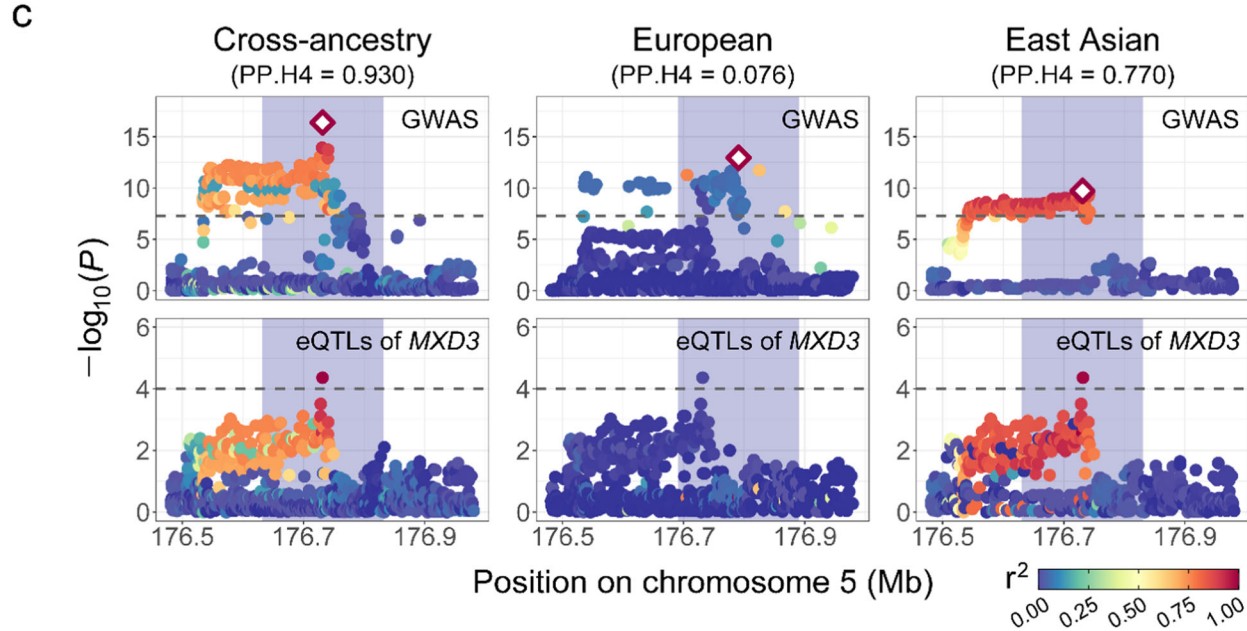

**Fig. 4 | Colocalization with eQTL in kidney tissues. a** Venn diagrams represent the number of colocalized genes in each study (top) and in each kidney tissue (bottom). **b** Comparison of the number of variants in the 95% credible set of each gene colocalized in each kidney tissue between the cross-ancestry (y-axis) and the European study (x-axis). Genes with a smaller number of variants in the credible set in the cross-ancestry study than those in the European study are lightly colored. **c** Regional plots (500 kb) of association analysis for serum urate (top) and *MXD3*

expression in the tubulointerstitial tissue (bottom). Each dot represents a variant plotted as -log$_{10}$ (P-value) on y-axis against the corresponding variant position (Mb) on the x-axis and variants are colored according to linkage disequilibrium with lead variants (rhombus) in each study. Blue shading (200 kb) is the region used for the colocalization analysis. P-values were determined using a two-sided test. GWAS genome-wide association study, eQTL expression quantitative trait loci.

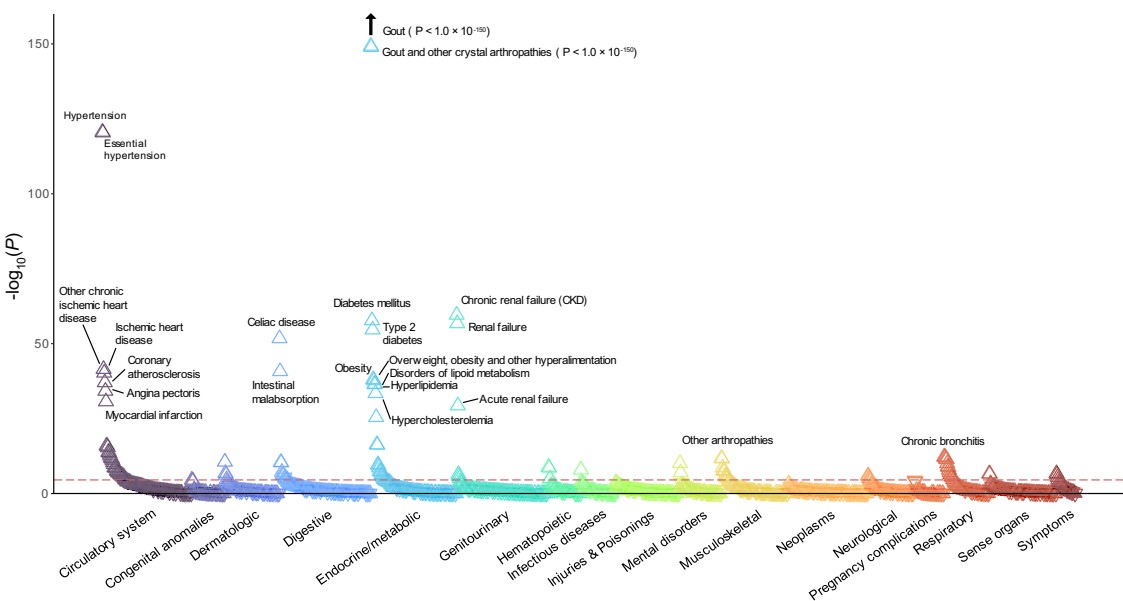

**Fig. 5 | Phenome-Wide Association Study (PheWAS) of cross-ancestry PRS in the UKBB participants of European ancestry.** PheWAS plot for 1621 UKBB phecodes using PRS derived from cross-ancestry meta-analysis results. The *x*-axis represents each phecode divided into 17 categories, and the *y*-axis represents the log$_{10}$ of uncorrected *P*-values for linear regression between SU PRS and each phecode. A triangular dot indicates a phenotype with a positive odds ratio, and an inverted triangular dot indicates a phenotype with a negative odds ratio. The upper dotted line represents the threshold for Bonferroni's correction ($P = 3.08 \times 10^{-5}$), and the lower dotted line represents a *P*-value of 0.05. PRS polygenic risk score, SU serum urate, UKBB UK BioBank.

respectively. The prevalence of gout was 1.31, 0.41, and 0.33% in the high, intermediate, and low groups, respectively (Supplementary Fig. 11 and Supplementary Data 22). This increasing pattern was also observed across the SU PRS decile groups. As for the odds ratio (OR) of each decile group, the PRS applied with the East Asian meta-analysis was generally higher for hypertension, and the PRS applied with the cross-ancestry meta-analysis was higher for gout, although the CIs overlapped (Supplementary Fig. 12 and Supplementary Data 23). The top PRS decile in East Asian ancestry had a 1.5-fold higher risk of hypertension, and the top PRS decile in cross-ancestry had a 7.1-fold higher risk of gout.

We constructed and evaluated a risk prediction model for each disease (Supplementary Figs. 13 and 14 and Supplementary Data 24). For both hypertension and gout, the combined model using the cross-ancestry meta-analysis PRS showed the best performance (area under the receiver operating characteristic curve = 0.718 and for hypertension, 0.793 for gout).

**Mendelian randomization analysis for causal inference**

To infer the causal relationships of SU with gout, heart failure, and hypertension, we used a two-sample Mendelian randomization (MR) approach. A putative causal effect of SU on gout and heart failure was detected using the inverse-variance weighted (IVW) regression MR test (gout, OR = 4.86, $P = 2.97 \times 10^{-36}$; heart failure, OR = 1.10, $P = 1.78 \times 10^{-4}$; hypertension, OR = 1.20, $P = 3.63 \times 10^{-6}$). The additional sensitivity tests had effects in the same direction as those of the IVW test. MR analysis was performed after pleiotropy correction by removing outlier variants (potentially pleiotropic variants) derived from MR-PRESSO v1.0 in hypertension. MR-Egger showed no evidence of horizontal pleiotropy (intercept = 0.003, $P = 0.085$) (Supplementary Data 25).

To identify genes that contribute to the causal relationship of SU with gout, heart failure, and hypertension, we conducted SMR v1.3.1 with 2671 SU-associated genes from enrichment analysis, colocalization analysis, and TWAS; 2263 and 2271 genes in the TUBE and GLOM tissues from the NEPTUNE study, respectively. A total of 467 and 323 genes in the TUBE and GLOM tissues, respectively, passed the nominal significance level (SMR $P < 0.05$) in the SMR analysis for SU as an

outcome (Supplementary Data 26). Among these genes, 13, 7, and 34 showed potential causal associations (FDR $P < 0.05$ and heterogeneity in dependent instruments (HEIDI) $P \geq 0.01$) with gout, heart failure, and hypertension, respectively, in the SMR analysis for each disease as an outcome (Supplementary Data 27). With these significant genes, we performed sensitivity analyses using other gene expression data: whole blood and kidney tissues from the GTEx v8[30] and blood tissue from the eQTLgen[31] datasets. We validated a concordant direction of effect sizes of *CTBP1*, *PPM1G*, *SEPT2*, and *KRTCAP3* for gout; *SKIV2L* for heart failure; and *AAK1*, *MAPKAPK5-AS1*, *POLA2*, *RSG1*, and *WWP2* for hypertension (Table 2). Among these ten genes, *SKIV2L* and *CTBP1* were identified only by analysis using cross-ancestry GWAS.

For the 10 genes validated in the sensitivity analyses, we investigated their indirect effects via SU using mediation analysis. The proportion of mediation effects of *CTBP1*, *PPM1G*, *SEPT2*, and *KRTCAP3* through SU on gout were 12.63, 12.82, 8.55, and 12.31%, respectively; that of *SKIV2L* through SU for heart failure was 3.34%; and that of *AAK1*, *MAPKAPK5-AS1*, *POLA2*, *RSG1*, and *WWP2* for hypertension were 7.75, 7.94, 6.26, 6.65, and 8.02%, respectively. Similar results were observed in the sensitivity analyses (Supplementary Data 28).

## Discussion

The aim of our study was to identify variants, genes, pathways, and traits associated and causally related to SU. We conducted a large-scale cross-ancestry meta-analysis of 460,894 individuals from the UKBB and 110,739 individuals from KoGES. This extended study included 1,029,323 individuals, which is approximately double the sample size of previous cross-ancestry studies. In this study, we identified 351 significant SU-associated genetic loci, including 17 previously unreported loci[13,32,33], which were more than 2 Mb away from the previously reported loci. We observed similar effect sizes for these loci between the European and East Asian populations. These SU-associated loci are enriched in the SU-related tissues, including the urinary tract and kidney in the urogenital system. Our GWAS meta-analysis provided additional insights that have not been thoroughly examined in previous SU studies. 1) The SU-associated loci showed similar effect sizes, high genetic correlation, and shared genetic architecture across

**Table 2 | Significant MR associations in both main and sensitivity analysis**

| Trait | Gene | eQTL source | eQTL tissue | Beta SMR | FDR P | P HEIDI |
|---|---|---|---|---|---|---|
| Main analysis | | | | | | |
| gout | *CTBP1* | NEPTUNE | GLOM | 0.081 | 6.53E−10 | 1.000 |
| gout | *PPM1G* | NEPTUNE | GLOM | −0.119 | 4.06E−06 | 0.202 |
| gout | *SETP2* | NEPTUNE | TUBULO | 0.146 | 3.87E−02 | 0.954 |
| gout | *KRTCAP3* | NEPTUNE | GLOM | 0.122 | 3.60E−04 | 0.510 |
| heart failure | *SKIV2L* | NEPTUNE | GLOM | 0.033 | 7.23E−04 | 0.879 |
| ESS HTN | *AAK1* | NEPTUNE | TUBULO | −0.023 | 1.27E−04 | 0.211 |
| ESS HTN | *MAPKAPK5-AS1* | NEPTUNE | TUBULO | 0.031 | 3.92E−04 | 0.399 |
| ESS HTN | *POLA2* | NEPTUNE | TUBULO | −0.028 | 8.85E−04 | 1.000 |
| ESS HTN | *RSG1* | NEPTUNE | TUBULO | 0.015 | 2.71E−03 | 1.000 |
| ESS HTN | *WWP2* | NEPTUNE | TUBULO | −0.039 | 2.14E−03 | 0.619 |
| Sensitivity analysis | | | | | | |
| gout | *CTBP1* | eQTLGEN | whole blood | 0.092 | 4.48E−02 | 0.034 |
| gout | *KRTCAP3* | GTEX v8 | whole blood | 0.121 | 2.21E−13 | 0.061 |
| gout | *PPM1G* | GTEX v8 | whole blood | −0.516 | 5.26E−05 | 0.743 |
| gout | *SETP2* | eQTLGEN | whole blood | 0.046 | 1.13E−02 | 1.000 |
| gout | *SETP2* | GTEX v8 | whole blood | 0.086 | 1.25E−02 | 1.000 |
| heart failure | *SKIV2L* | GTEX v8 | whole blood | 0.027 | 1.08E−02 | 0.970 |
| ESS HTN | *AAK1* | eQTLGEN | whole blood | −0.153 | 6.27E−03 | 1.000 |
| ESS HTN | *AAK1* | GTEX v8 | whole blood | −0.019 | 1.59E−03 | 0.658 |
| ESS HTN | *MAPKAPK5-AS1* | GTEX v8 | whole blood | 0.033 | 5.54E−13 | 0.014 |
| ESS HTN | *POLA2* | eQTLGEN | whole blood | −0.027 | 5.51E−97 | 0.151 |
| ESS HTN | *POLA2* | GTEX v8 | whole blood | −0.035 | 4.62E−16 | 0.414 |
| ESS HTN | *RSG1* | GTEX v8 | whole blood | 0.026 | 8.14E−04 | 0.998 |
| ESS HTN | *WWP2* | GTEX v8 | whole blood | −0.060 | 4.78E−10 | 0.023 |

Significant MR associations were observed in both main analysis using NEPTUNE kidney tissue and sensitivity analysis using eQTLgen, GTEx v8 whole blood and kidney tissue. Results with opposite beta SMR between the main analysis and sensitivity analysis were excluded. Significant MR associations that passed the FDR correction (FDR *P* ≤ 0.05). *P*-values were determined using a two-sided test.

*MR* Mendelian randomization, *ESS HTN* essential hypertension, *GLOM* glomerular, *TUBULO* tubulointerstitial, *CHR* chromosome, *BP* base position, *SNP* single nucleotide polymorphism, *eQTL* expression quantitative trait loci, *A1* effective allele, *SMR* summary-based Mendelian randomization, *FDR P* *P*-value adjusted by false discovery rate, *HEIDI* heterogeneity in dependent instruments.

ancestries, which was in line with the findings on the traits from other studies[34,35]. The effect sizes of the lead variants were positively correlated ($\rho = 0.426-0.77$) across the European, East Asian, and other ancestries. 2) In addition to the cross-ancestry analysis, we conducted downstream analyses based on the GWAS results for each ancestry, which allowed some analyses to identify ancestry-specific results. 3) We identified 467 and 323 potential causal genes in the tubulointerstitial and glomerular kidney tissues, respectively, among 2671 genes in the SU-associated GWAS loci, through a series of enrichment analysis, colocalization analysis, TWAS, and SMR. 4) In the PheWAS with PRS, the PRS of SU was significantly associated with gout, heart failure, and hypertension. We identified the significant potential causal effects of SU on these SU-associated diseases, including heart failure and hypertension, which was previously controversial. 5) We identified ten genes that showed potential causal associations of SU along with heart failure and hypertension and investigated their effects on the diseases through SU.

Colocalization analysis identified 173 potentially causal genes in the SU-associated loci, including 36 genes identified in the ancestry-specific analysis. Most of the colocalized genes had a smaller credible set size in the cross-ancestry than in the European cohort, and more genes were colocalized in cross-ancestry. As shown with *MXD3*, cross-ancestry analysis helped identify potentially causal genes with greater power than ancestry-specific analyses and averaged linkage disequilibrium (LD) patterns across ancestries. Potentially causal genes within a particular ethnic group can be identified in ancestry-specific analysis, but methods that more delicately consider LD patterns are

required. The eQTL data for colocalization analysis were based on gene expression levels in patients with nephrotic syndrome (NEPTUNE study), which might affect the colocalization results of SU-associated variants in the general population.

This study provides valuable insights into the genetics and biology of SU and its related diseases. The cross-ancestry meta-analysis results were enriched in most tissues, including the urinary tract of the urogenital system, kidney, and cartilage and exhibited the lowest *P*-values. In addition, we observed ancestry-specific enrichment in tissues such as the cardiovascular tissues in European ancestry and the fetal blood and the nasal mucosa tissues in East Asian ancestry. The enrichment in the cardiovascular tissues was consistent with the results of the MR analysis of the European ancestry, which identified a potential causal relationship between SU and both heart failure and hypertension. The East Asian meta-analysis results were enriched in the fetal blood of the hematologic and immune system ($P = 0.001$) and the nasal mucosa of the respiratory system ($P = 3.65 \times 10^{-3}$). Previous clinical studies have shown that SU is associated with fetal growth[36,37], and studies on the association between various air pollutants and the nasal cavity have revealed urate as an important first-line defense factor against reactive oxygen species[38,39]. Moreover, both ancestry-specific GWAS results were enriched in digestive system tissues. The association between SU and the intestinal tract is consistent with previous studies that found that intestinal *ABCG2* dysfunction was a cause of hyperuricemia[40,41]. Gene set enrichment analysis identified that the systemic lupus erythematosus and TGF-β signaling pathways were associated with SU in cross-ancestry and European ancestry. SU is a risk

predictor or therapeutic factor for systemic lupus erythematosus[27,42]. Previous experimental studies showed that a decrease in SU had a preventive effect against TGF-β1-induced profibrogenic progression in patients with type 2 diabetic kidney disease[43]. Cross-ancestry analysis identified an association between the extracellular matrix receptor interaction gene set and SU, consistent with a previous study reporting elevated SU in renal fibrosis and offend-stage chronic kidney disease (CKD)[44]. SU GWAS was also enriched in the focal adhesion gene set that included the *IBSP* gene, related to vascular calcification and a strong prognostic marker for cardiovascular mortality in CKD patients[45,46]. Only the East Asian ancestry analysis revealed an association between SU and the MAPK signaling pathway, supported by previous findings that SU is associated with renal tissue growth through the MAPK pathway[47]. The ancestry-specific findings in this study have two possible explanations. It is possible that ancestry-specific genetic loci affect SU-related biological pathways in certain ancestral populations only, or that the identification of such unique loci and the subsequent findings based on them may also be due to the differences in statistical power in each ancestry. For example, despite shared biological mechanisms across ancestries, some genetic loci can be identified as ancestry-specific loci that are unidentifiable in other ancestries or cross-ancestry meta-analysis, owing to several factors, such as different allele frequencies, LD structure, and environmental factors. Therefore, the interpretation of ancestry-specific findings requires caution, and comparisons are warranted for larger datasets across ancestries. Nevertheless, genetic studies of diverse ancestries and the findings from each ancestry may provide new and valuable insights into the biological background of SU.

Cross-ancestry PRS had advantages over European and East Asian ancestry PRSs. PRS PheWAS performed in the UKBB European population, applying cross-ancestry and European PRSs, found >80% of significant phenotypes in both the cross-ancestry and European PRS analyses had higher R-squared values in the European PRS. When cross-ancestry PRS was applied, new associated phenotypes were found. Erythematous conditions ($P = 2.07 \times 10^{-5}$) are symptoms of acute gout[48], and pulmonary heart disease was associated with SU in previous clinical studies[49,50]. As other alveolar and parietoalveolar pneumonopathies are significantly associated with SU PRS, this should be further studied in clinical studies of SU and lung-related diseases[51,52]. Consistent with the results from the European population, gout and hypertension were significantly associated with the SU PRS in the PheWAS of the Korean population. The predictive utility of the ancestry-specific SU PRS for gout, heart failure, and hypertension was evaluated through survival analysis in the UKBB European population. Zhang et al.[53] reported that individuals with high gout PRS (highest tertile) had a 77% higher risk of gout than those with low genetic risk (lowest tertile). Although the improvement of gout risk prediction by SU PRS has been previously examined[13], no studies have examined the association of SU PRS with the risk of heart failure and hypertension. This study found that individuals with very high (99th percentile) SU cross-ancestry PRS had 7.00-, 1.37-, and 1.34-fold higher risks of gout, heart failure, and hypertension, respectively, than those with low (0–19th percentile) SU cross-ancestry PRS. The association between SU and hypertension, shown in previous observational studies, was also confirmed using PRS in the East Asian population[54]. These results suggest that PRS can help identify individuals with a high genetic predisposition to specific diseases associated with SU, though the predictive ability should be improved.

SU has been extensively studied, with most studies confirming its direct causative role in gout. However, its role in other diseases remains controversial. Stewart et al. highlighted the difficulty in inferring a direct causal relationship between hyperuricemia and hypertension and suggested that large-scale randomized trials are required to further elucidate this relationship[55]. A study by Krishnan et al. was pivotal in identifying SU as a potential risk factor for heart failure, which had previously been unrecognized[56]. However, large-scale MR studies have failed to identify a causal relationship among SU, blood pressure, and heart failure[57,58]. In addition, umbrella reviews of SU have failed to find convincing evidence regarding the clear role of SU in diseases other than gout and nephrolithiasis[1]. In contrast to these negative findings, a recent MR study showed that genetically determined SU levels were significantly associated with heart failure (OR = 1.07, 95% CI = 1.03–1.10; $P = 8.6 \times 10^{-5}$)[59]. The current study replicated this result using more instrumental variables (OR = 1.10; 95% CI = 1.05–1.16; $P = 1.78 \times 10^{-4}$). Although associations between SU and hypertension have been reported[60,61], a causal genetic association has not yet been established in MR research[57,62]. Our study demonstrated a potential causal association between SU and hypertension without genetic pleiotropy (OR = 1.20; 95% CI = 1.11–1.30; $P = 2.80 \times 10^{-6}$); this is in contrast to a previous report about the existence of genetic pleiotropy[63]. This result provided evidence that a direct causal relationship may exist without the pleiotropic effect of SU on hypertension, which is consistent with previous studies that suggested a causal relationship between SU and BP and gout with hypertension, respectively[64,65].

For gout, heart failure, and hypertension, which were potentially causally associated with SU, SMR analysis was performed to infer the association between these traits and the expression of 2671 SU-related genes selected from the enrichment analysis, colocalization analysis, and TWAS. The associations between these genes and traits were investigated to identify new treatment targets for these three diseases. Typically, the ULT methods employed to reduce SU levels involve the use of drugs with specific mechanisms of action (MOA), such as xanthine oxidase inhibitors (XOIs), uricosuric agents, and uricase. The 2020 ACR guidelines recommends XOIs as the first-line treatment for patients with gout[5]. However, only two drugs (allopurinol and febuxostat) are widely used as XOIs. Several large randomized clinical trials have shown that allopurinol is ineffective in the treatment of hypertension, CKD, and ischemic heart disease[66–70]. This indicates that ULT drugs are almost exclusively effective in the treatment of gout. A review of all currently available ULTs highlights the need for new ULTs with multiple mechanisms[6]. Our study identified candidate genes for new ULT methods (four for gout, one for heart failure, and five for hypertension) that may have a direct causal effect on SU and an indirect effect on the three diseases via SU. For these genes, the proportion of the mediation effect of SU on gout, heart failure, and hypertension was examined and found to be smaller than that of the direct effect. These genes may have multiple mechanisms in these three diseases, including direct and indirect effects via SU. Further research is required to elucidate the functions of these genes.

Although a direct association between these genes and SU has not been reported, this potential relationship is supported by other biological experimental studies. We investigated the MOA of the ten ULT candidate genes identified in our study. *SKIV2L* is one of the complexes that make up the RNA exosome, and its various roles in the exosome have emerged[71,72]. The *SKIV2L* exosome is closely related to the immune response[73] and may be causally associated with heart failure via immune system activation[74,75]. *CTBP1* is a co-repressor complex in the notch signal pathway, and activation of the notch signal due to an increase in urate levels leads to an inflammatory response that causes gout[76–79]. Although little is known about the function of *KRTCAP3*, it affects obesity and insulin sensitivity[80]. Previous studies have demonstrated that increased urate is associated with the risk of developing diabetic nephropathy in diabetic patients, and gout is associated with diabetes[81,82]. *WWP2* is a member of the Nedd4 family of E3 ligases, which plays an important role in protein ubiquitination. *WWP2* is involved in endothelial injury and vascular remodeling after endothelial injury as a novel regulatory factor, suggesting a possible new target for the prevention and treatment of hypertensive vascular disease[83]. Although a direct functional relationship between these

genes and SU remains unclear, further studies are required to identify the precise biological mechanisms underlying these potential target genes.

In summary, we investigated variants, genes, tissues, pathways, and diseases associated with SU and potential therapeutic targets through the largest cross-ancestry and ancestry-specific meta-analysis for SU to date. This approach highlighted the potential of repositioned drugs targeting SU for the treatment of other diseases. In addition, we identified potential causal relationships between SU, target genes, and various diseases. Our study further adds insight into the genetic architecture by leveraging abundant genomic resources.

## Methods

### Characteristic of study cohorts
We performed a meta-analysis using six summary statistics for the four cohorts. First, we produced summary statistics for serum urate (SU) using genotype and phenotype data from the UKBB (UK Biobank) database (release version 2). UKBB is a large-scale biomedical database and research resource containing in-depth genetic and health information from half a million participants in the United Kingdom. Of approximately 500,000 samples, 460,894 individuals with information on SU were selected, and analysis was performed by dividing them into 388,724 Europeans and 72,170 non-Europeans using genetic ethnic grouping (UKBB field ID 22006) information. Second, we used the summary statistics of the Chronic Kidney Disease Genetics Consortium (CKDGen), a cross-ancestry study that meta-analyzed 74 multiple-ancestry SU studies[13]. Summary statistics for the cross-ancestry GWAS (457,690 individuals) and the European GWAS (288,649 individuals) were provided by the CKDGen, and the individuals in both datasets were not included in the UKBB. Third, we used the analysis of East Asian ancestry GWAS SU summary statistics of 110,739 individuals from the Korean Genome and Epidemiology Study (KoGES) cohort provided by the South Korea National Institute of Health. Additionally, 22,607 individuals, independent of the 110,739 individuals involved in the discovery analysis, were included in the replicates. Fourth, SU GWAS summary statistics of 109,029 individuals provided by the BioBank Japan Project were used[32]. This cohort was analyzed by classifying it as East Asian ancestry. Detailed information about each cohort is presented in Supplementary Data 1. The genotype data and summary statistics of all cohorts used in the meta-analysis were aligned with the Genome Reference Consortium human build (GRCh) 37.

### Genotype data quality control and GWAS
UKBB (release version 2) performed quality control (QC) and GWAS after removing sex mismatch and aneuploidy samples and dividing them into European and non-European groups according to genetic ethnic grouping (Data-Field 22006). Kinship was not removed among all individuals, and variant QC was performed with a call rate <0.95, minor allele frequency (MAF) < 0.005, Hardy-Weinberg equilibrium (HWE) $P < 1.0 \times 10^{-6}$, and INFO < 0.4. Association analysis was performed using linear mixed model analysis with BOLT-LMM v2.3.4[84] as the residual value of SU (mg/dl) ≈ AGE + SEX. The first four principal components (PCs) of genetic ancestry, which were calculated based on the entire UKBB population provided by the UKBB (data field 22009), were used as covariates. To examine the robustness of the PCs in accounting for population stratification, we additionally performed a PCA using HapMap phase 3 variants on the unrelated European individuals ($N = 276,250$) from the UKBB who self-identified as 'White British' (data-field 21000) and have very similar genetic ancestry based on a PCA of the genotypes (data-field 22006). We then calculated the PCs of all UKBB European individuals ($N = 408,188$) using PC loadings from the PCA. The Spearman's correlation coefficient was used to compare the beta coefficients from the GWAS of the two sets of PCs (Supplementary Fig. 2). To examine population stratification in the GWAS of UKBB non-European individuals, we defined seven

genetically distinct groups for individuals categorized as having non-White British" ancestry based on the PC values provided by UKBB, as delineated by Privé et al.[85]. We performed a GWAS of SU in each of the seven genetically distinct groups separately and analyzed the results. Association analysis was performed using linear mixed model analysis with SAIGE v1.1.3[86] as the residual value of SU (mg/dl) ≈ AGE + SEX. The first four PCs of genetic ancestry, calculated for each ancestry group using PLINK v2.0[87], were used as covariates. The meta-analysis of seven non-European GWAS ($N = 28,320$) showed highly consistent effect sizes of the analyzed variants, but slightly less significant associations in comparison with the combined non-European GWAS adjusted for PCs provided by the UKBB ($N = 72,170$) (Supplementary Fig. 3). Based on this observation, the analyses in this study were conducted using 72,170 non-European individuals from the UKBB to enhance the statistical power.

In the case of the KoGES cohort, the Korea Biobank Array (KBA) project genotyped individuals from three population-based cohorts, a part of KoGES, using the KBA, a customized genotyping array optimized for Korean genome research. The three cohorts were Ansung-Ansan, health examinee, and cardiovascular disease association study. QC of the genotype data was performed with the following criteria[17,88]. Briefly, genotypes were called per batch (3–8 K samples) considering recruitment year and site. In the subsequent sample QC, putative low-quality samples were removed if sex inconsistency, low call rate (<97%), excessive heterozygosity, and outliers from the principal component analysis (PCA) result. Additionally, samples with second-degree relatedness were removed using KING v.2[89]. In variant QC, variants were excluded if call rate <0.95 and HWE $P < 1.0 \times 10^{-6}$. QCed genotypes were phased using Eagle v2.3[90] and the following imputation analysis was performed using IMPUTE v.4[91] with a merged reference panel from 2504 samples of the 1000 Genomes Project phase 3 and 397 samples of the Korean Reference Genome. After imputation, variants with INFO less than 0.8 or MAF < 1% were removed. Prior to association analysis, the level of SU was transformed by taking the residuals from the following equation: SU (mg/dl) ≈ AGE + SEX. Single variant association analysis was performed on the residuals adjusting for four PCs using the EPACTS package v.3.2.6 [See URL: https://genome.sph.umich.edu/wiki/EPACTS].

### Ancestry-specific meta-analysis
After checking each GWAS summary statistic using GWAtoolbox v2.2.4-10[92] and custom scripts, the A1 allele was set the same between cohorts. The QC levels of the variants were adjusted equally to MAF > 0.005 and INFO > 0.6. Cross-ancestry meta-analysis was performed on UKBB (European and non-European), CKDgen (cross-ancestry), and KoGES (East Asian). For European ancestry, a meta-analysis of UKBB (European) and CKDgen (European) was performed. For East Asian ancestry, a meta-analysis of BioBank Japan (BBJ, East Asian) and KoGES (East Asian) was performed. The meta-analyses were performed using a fixed-effect inverse-variance weighted (IVW) meta-analysis using METAL (released on 2011-03-25)[93]. When performing meta-analysis for each ancestry, only the variants common to at least half of the cohorts included in the analysis were extracted. BBJ was not added in the cross-ancestry meta-analysis to avoid data duplication because CKDgen (cross-ancestry) already included BBJ.

We changed the $P$-value by finding the lowest value that was not recognized as zero for each ancestry-specific meta-analysis to prevent cases in which the $P$-value was recognized as zero and excluded from other analyses.

### Significance criteria for GWAS loci
Lead loci were similarly extracted from the results of each ancestry-specific meta-analysis. The SNP with the smallest $P$-value for each chromosome was selected as the lead locus. Based on these loci, the flanking 500 kb was considered as one region (1 Mb). Regions of this

size were extracted until no more significant loci ($P < 5.0 \times 10^{-8}$) were found across the entire genome. Therefore, the lead loci were at least 500 kb apart. Functional annotation of the lead loci for each ancestry was performed using ANNOVAR (released 2019-09-27)[94].

For set-specific significant loci, loci unique to each ancestry meta-analysis were selected by comparing the lead variants extracted from each ancestry-specific meta-analysis. Based on the lead variants of each ancestry meta-analysis, set-specific significant loci were selected by comparing significant lead variants between ancestry meta-analyses ($P < 5.0 \times 10^{-8}$) and by confirming that lead variants in one ancestry-specific analysis were more than 1 Mb away from those in the other ancestry-specific analysis.

In the cross-ancestry meta-analysis, previously unreported loci were identified. These loci were compared with the reported loci from the GWAS Catalog (ver. 22 June 2023) and recently published SU GWASs[95–97]. To define unreported loci strictly, we regarded our significant loci as previously unreported loci if there were no significant variants ($P < 5.0 \times 10^{-8}$) in other SU GWASs in the 2 Mb (4 Mb region) window on both sides from our lead SNPs.

Regional plots for 17 previously unreported significant loci were generated using LocusZoom v1.4[98]. Linkage disequilibrium (LD) information was calculated and used from the 1000 Genomes Project Phase 3 data.

## Comparison of effect size of ancestry-specific lead loci

The effect size was compared by extracting the common variants from the European ancestry and the East Asian ancestry meta-analysis with the lead loci ($P < 5.0 \times 10^{-8}$) of each of the three ancestry-specific meta-analyses. Out of 351 cross-ancestry meta-analysis lead loci ($P < 5.0 \times 10^{-8}$), only 263 variants common in the European and East Asian ancestry meta-analyses were extracted to compare the effect size (beta coefficient). In the case of 269 lead loci ($P < 5.0 \times 10^{-8}$) of the European ancestry analysis, the effect size was compared with 190 variants common to the East Asian ancestry analysis. In the case of 90 East Asian lead loci ($P < 5.0 \times 10^{-8}$), 65 variants commonly present in the European ancestry analysis were compared. Spearman's correlation test and Cohen's kappa coefficient were used to compare effect sizes and investigate the directional consistency of genetic effects between European and East Asian ancestry analyses.

We compared the effect sizes of the lead variants in the cross-ancestry GWAS meta-analysis with those from the GWAS of each of the four genetically distinct groups in UKBB with a sample size of >3000 individuals. The GWAS was performed for each group using the same QC process that was used for the UKBB European GWAS. Among the lead variants in the cross-ancestry GWAS meta-analysis, 263, 323, 177, and 331 variants were found in the data from India, Italy, Nigeria, and Poland, respectively. Spearman's correlation and Cohen's kappa coefficients were used to compare the effect sizes and directional consistencies of the genetic effects, respectively.

## Genetic heritability and genetic correlation in European and East Asian ancestry, and genetic correlation between the two ancestries

The single nucleotide polymorphism (SNP)-based heritability of each ancestry-specific meta-analysis was calculated using LD score regression (LDSC) v1.0.1[25]. After meta-analysis, we adjusted the value to the lowest possible level in the software to prevent variants with a too low $P$-value from being excluded from the calculation. For quantification of the explanatory power of cross-ancestry meta-analysis, the proportion of SU variance explained by lead SNPs was calculated by referring to Tin et al.; $\beta^2(\frac{2p(1-p)}{var})$, where $\beta$ is the effect size for SU, $p$ is the minor allele frequency, and var is the phenotypic variance.

Furthermore, we calculated genetic correlations with 320 other traits of European ancestry and 181 other traits of East Asian ancestry using public GWAS data. Significant results were obtained through

FDR correction within the genetic correlation results for each ancestry (FDR < 0.05). The pre-calculated LD score for each ancestry was used by receiving data based on the 1000 Genome Project phase 3 from the LDSC website.

The cross-population genetic effect correlation between European and East Asian ancestry was performed using Popcorn[19]. This software can be used to obtain two types of common-SNP-based cross-population genetic correlations. These are genetic effect correlation and genetic impact correlation; among them, genetic effect correlation is known as a more realistic model. Pre-computed scores for the European and East Asian 1000 Genomes Project data were downloaded from the software website.

## Functional enrichment analysis for GWAS loci

We performed functional enrichment analysis for each ancestry-specific meta-analysis. Tissue enrichment analysis was performed using Data-driven Expression Prioritized Integration for Complex Traits (DEPICT)[99]. Among the SU-associated variants for each ancestry, those with $P$-value $< 1.0 \times 10^{-5}$ were used as input. Independent SNPs were identified using the PLINK v1.9[100] clump command within 500 kb flanking regions and $r^2 > 0.2$ in the 1000 Genomes project phase 3 data for each ancestry. Significant tissue ($q < 0.05$) was separately indicated through false discovery rate (FDR) correction.

Gene set enrichment analysis was performed using GSA-SNP2 (released 2020-09-01)[101]. The padding size was set to 20 kb so that genes with a high correlation adjacent to the SNP could be included as much as possible. Gene set annotation used 2,982 canonical pathway gene sets (KEGG, REACTOME, and NABA) among curated gene sets of the MSigDB c2.cp.v6.2 database[102]. Associated gene sets were selected only if they passed the FDR threshold ($q < 0.25$).

## Colocalization with expression quantitative trait locus from NEPTUNE

We performed colocalization between ancestry-specific meta-analysis and *cis*- expression quantitative trait locus (eQTL) results in micro-dissected human glomerular (from 240 individuals) and tubulointerstitial (from 311 individuals) kidney tissues from the Nephrotic Syndrome Rare Disease Clinical Research Network III (NEPTUNE)[28] using the coloc.abf function from the R package coloc v5.1.1[103]. The colocalization analysis was conducted for loci within ±100 kb of each lead variant from ancestry-specific meta-analyses. Among the association pairs of each locus, those containing less than 30 *cis*-eQTLs or having a minimum eQTL $P$-value $> 1.0 \times 10^{-4}$ were excluded from analysis. European, East Asian, and European + East Asian samples from the 1000 Genomes Project phase 3[104] were used as LD reference panels for the European, East Asian, and cross-ancestry meta-analyses, respectively. Association pairs with a posterior probability for colocalization (PP.H4) greater than 0.8 were considered colocalized. Variant-level posterior probabilities for colocalization (SNP.PP.H4) were derived from the colocalization analysis. The 95% credible set represented the smallest set of SNPs with a cumulative SNP.PP.H4 greater than 95%.

## Transcriptome-wide association study

Transcriptome-wide association study (TWAS) analysis is a method used to determine the association between the expression of the transcriptome and a specific trait, and genes that are different from those mapped by GWAS analysis are selected. We conducted a TWAS to determine the relevant eQTL for each ancestry-specific meta-analysis using PrediXcan v0.7.5[29]. A total of 49 GTEx v8 tissue eQTL data of the MASHR-based model provided by PrediXcan were used to find related genes for each tissue. The most significantly associated gene was found by integrating all tissues using the S-multiXcan function. Genes that passed the Bonferroni correction were considered significant ($P_{bon} = 0.05/21,681$ in cross-ancestry, $P_{bon} = 0.05/21,681$ in

Europe, $P_{bon}$ = 0.05/20,154 in East Asia), and these genes were used for the eQTL Mendelian randomization analysis.

## Leave-one-out polygenic risk score

Tenfold leave-one-out PRS (LOO PRS) was performed for each ancestry-specific genotype. In the case of European ancestry, the UKBB data were randomly divided into ten equal parts, and association analysis was performed on nine datasets using BOLT-LMM in the same way as described above. Then, cross-ancestry and European meta-analyses were performed with METAL (released on 2011-03-25), and PRS was calculated by applying these summary statistics to the remaining UKBB data. Thus, cross-ancestry and European PRS for UKBB individuals were obtained. This process was repeated ten times to calculate the PRS for all samples of the UKBB data, so that sample overlap did not occur. The Pearson correlation coefficient ($R^2$) between PRS and SU residuals calculated in this way was 0.306 and 0.310 for the cross-ancestry and European PRS, respectively.

Similarly, we performed LOO PRS for the East Asian ancestry. The KoGES data of 72,299 of the 110,739 unrelated individuals, whose individual-level genotype data were available in this study, were randomly divided into ten groups. Association analysis was performed on nine datasets using PLINK v1.9, and the PRS was calculated for the remaining dataset. As a result, cross-ancestry and East Asian PRS for 72,299 individuals from the KoGES were obtained. The Pearson correlation coefficients between the PRS and SU residuals were 0.272 and 0.267 for the cross-ancestry and East Asian PRS, respectively.

PRS calculations were performed using PRS-CS-auto (released on 2021-06-04)[105]. An additional detailed explanation of this process (Supplementary Fig. 8).

## Disease risk prediction in the replication study using PRS from the cross-ancestry and East Asian ancestry

We conducted disease risk prediction using the cross-ancestry and East Asian ancestry PRS in an independent dataset comprising 22,607 Korean individuals from the KoGES. This dataset was independent from the 110,739 KoGES individuals used in the discovery analysis. The QC procedures for the replication dataset have been described elsewhere[17]. To avoid overfitting, we constructed adjusted weights for PRS analysis using summary statistics from the discovery studies without the replication dataset (Supplementary Data 1). We obtained each ancestry-specific PRS by applying previously performed cross-ancestry and East Asian ancestry meta-analyses. The target diseases, hypertension and gout, were defined based on self-reports.

1. We assessed the prevalence of hypertension and gout in the PRS group. The PRS was divided into quartiles, and the second and third quartiles were grouped as "intermediate" to compare the lower 25, middle 50, and upper 25%.

2. We partitioned the individuals into PRS deciles and estimated the OR between the first decile and each of the other decile groups.

3. Logistic regression was performed for hypertension and gout to test the performance of PRS by ancestry. ROC curves for PRS, demographic (AGE + SEX) + 4 PCs, and combined (PRS + AGE + SEX + 4 PCs) for each disease were calculated using pROC ver.1.18.0 in R[106].

## Phenome-wide association study in European and East Asian populations using cross-ancestry and ancestry-specific PRS

Cross-ancestry and European ancestry PRS for UKBB individuals obtained from the LOO PRS were adjusted by age, sex, and four PCs. The phenome-wide association study (pheWAS) was performed using Firth's bias-reduced logistic regression model for 1621 UKBB phecodes. For multiple-comparison correction, a phenotype significantly related to SU PRS was derived by reducing false-positive results using the Bonferroni correction method ($P_{bon}$ = 0.05/1621). Among these results, the phenotype showing a very significant level was used for Mendelian randomization analysis. We conducted the PheWAS with PRS in an East

Asian population using a similar approach. Cross-ancestry and East Asian PRS for KoGES individuals obtained from the LOO PRS were adjusted for age, sex, and the first four PCs. PheWAS was performed using Firth's bias-reduced logistic regression model for 37 self-reported diseases in the KoGES. We considered results with P-values less than the Bonferroni correction threshold ($P_{bon}$ = 0.05/37) to be significant.

We commonly applied $1.0 \times 10^{-317}$, the lowest value that can be expressed when the result value was zero.

## Survival test in European ancestry using cross-ancestry and European-ancestry PRS

We used UKBB individuals and cross-ancestry and European ancestry PRS for the survival test. We excluded UKBB individuals who took ULT-related medications at enrollment (data field 20003; allopurinol and probenecid) from the survival analyses. The baseline characteristics of the study population were compared by SU PRS group using ANOVA for continuous variables and chi-squared tests for categorical variables. The follow-up year was used as the time scale in the model. The follow-up time was calculated from baseline assessment until the first event (gout, heart failure, and hypertension), death, or February 31, 2021, whichever occurred first. Essential hypertension, defined by ICD code (I10) was analyzed.

Cox proportional hazards regression models were applied to estimate the hazard ratio (HR) and 95% confidence interval (95% CI) of gout or heart failure concerning the genetic risk of SU adjusted for sex, age, and four PCs. The HR of SU PRS for gout, heart failure, and hypertension were used both as quantitative variables reported per one standard deviation and as categorical variables defined as follows: low (0–19th percentile), intermediate (20–79th percentile), high (80–98th percentile), and very high (99th percentile). All analyses were performed using the survival and survplot package in R. All P values were two-sided, and the statistical significance threshold was set at 0.05.

## Two-sample Mendelian randomization in European population

For Mendelian randomization (MR) analyses, we used GWAS results from a meta-analysis of European ancestry (CKDgen and UKBB) for SU, the Global Urate Genetics Consortium (GUGC) for gout[33], the Heart Failure Molecular Epidemiology for Therapeutic Targets (HERMES) Consortium for heart failure[107], and FinnGen for hypertension[108]. We performed two-sample MR (TSMR) using the MR-Base and 'TwoSampleMR' v0.5.6 package in R[109]. We used conventional IVW MR analysis as a principal MR test. We also conducted MR-Egger, simple mode, weighted median, and weighted mode methods as a sensitivity analysis, which are more robust to potential violations of standard instrumental variable assumptions. MR-PRESSO analysis was performed with the number of bootstrap replications of 100,000 times for pleiotropy correction and identification of potentially pleiotropic variants. MR-Egger intercept test was conducted to check the presence of potential pleiotropy. In every MR analysis, we clumped genome-wide significance ($P < 5 \times 10^{-8}$) SNPs with $r^2$ values ≤ 0.001. FinnGen GWAS summary statistics for essential hypertension defined by the ICD code (I10) were used.

## Summary-data-based Mendelian randomization in the European population

We used summary-based Mendelian randomization (SMR) v1.3.1 to determine associations between the expression of SU-associated genes and gout, heart failure, and hypertension[110]. We gathered SU-associated genes from DEPICT, TWAS, and colocalization analysis and then selected the same genes that had expression data in the NEPTUNE kidney tissue. In the main analysis, we performed SMR analysis with kidney gene expression (NEPTUNE tubulointerstitial and glomerular tissue) using European-specific SU GWAS. Genes whose expression in

the kidney was associated with SU ($P < 0.05$) were included in further SMR analysis with gout GWAS data from the GUGC, heart failure GWAS from HERMES, and hypertension GWAS from FinnGen. FDR $P$-value < 0.05, and HEIDI $P$-value > 0.01, were used for determining association and distinguishing pleiotropic associations from LD. For significant SMR associations, we performed additional sensitivity analysis with other expression data: GTEx whole blood tissue, GTEx kidney tissue, and eQTLgen blood tissue. Based on the MR results, which indicated that high SU levels were associated with an increased risk of gout, heart failure, and hypertension, we excluded the genes that showed opposite effects on SU and disease.

For proteins that causally associate in both the main and sensitivity analysis, we conducted a mediation analysis to estimate the effects of proteins on traits via SU. The "total" effect of protein on trait and effects of protein on SU were utilized with the previous SMR analysis. The effects of SU on traits were captured by the previous MR analysis. We used the product method and the delta method to estimate the beta, standard error, and confidence interval of the indirect effect.

### Reporting summary
Further information on research design is available in the Nature Portfolio Reporting Summary linked to this article.

## Data availability
The full summary statistics of cross-ancestry, East Asian, and European GWAS are publicly available at the NHGRI-EBI GWAS Catalog (https://www.ebi.ac.uk/gwas/downloads) with accession numbers GCST90319904, GCST90319905, and GCST90319906, respectively. The UKBB genotype and epidemiologic data are available by requesting access on the UKBB homepage (https://www.ukbiobank.ac.uk/). Summary statistics are publicly available from the Chronic Kidney Disease Genetics Consortium (CKDGen, http://ckdgen.imbi.uni-freiburg.de/). BBJ summary statistics were downloaded from the Biobank Japan PheWeb (https://pheweb.jp/). The full summary statistics of the KBA GWAS are available at the NHGRI-EBI GWAS Catalog (https://www.ebi.ac.uk/gwas/downloads) and the Korea National Institute of Health PheWeb (https://coda.nih.go.kr/usab/pheweb/intro.do). The GTEx data are publicly available upon reasonable application (http://www.gtexportal.org/home/datasets). The NEPTUNE eQTL data are publicly available (https://nephqtl.org). The HERMES GWAS summary statistics are publicly available (https://www.hermesconsortium.org). The FinnGen GWAS summary statistics are publicly available (https://www.finngen.fi/en). The GUGC GWAS summary statistics are publicly available (https://kp4cd.org/node/179).

## Code availability
Previously developed pipelines were used to produce the results for the current study. No custom code was developed. Please see the Supplementary Information for details on the software URLs and data used.

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

## Acknowledgements

This work was supported by a National Research Foundation of Korea grant funded by the Ministry of Science and Information and Communication Technologies, South Korea (2022R1A2C2009998) and an intramural grant from the Korea National Institute of Health (2022-NI-067-00). KBA data were provided by the Collaborative Genome Program for Fostering New Post-Genome Industry (3000–3031b). This study was conducted using bioresources from the National Biobank of Korea, the Korea Disease Control and Prevention Agency, Republic of Korea (NBK-2019-063). UKBB data were obtained under application no. 33002.

## Author contributions

H.H.Won and Y.J.Kim had full access to all data in the study and took responsibility for the integrity of the data and accuracy of the data analysis. C.Cho, B.Kim, and H.H.Won conceived and designed the study. C.Cho, B.Kim, and D.S.Kim performed the statistical analyses. C.Cho,

B.Kim, D.S.Kim, M.Y.Hwang, and H.H.Won drafted the manuscript. H.H.Won and Y.J.Kim supervised the study. C.Cho, B.Kim, and D.S.Kim contributed equally to this study. C.Cho, B.Kim, D.S.Kim, M.Y.Hwang, I.Shim, M.Song, Y.C.Lee, S.H.Jung, S.K.Cho, W.Y.Park, W.Myung, B.J.Kim, R.Do, H.K.Choi, T.R.Merriman, Y.J.Kim, and H.H.Won contributed to the interpretation of the data, writing the manuscript, and have read and approved the final draft for submission.

## Competing interests

W.-Y.P. is an employee of Geninus though this is unrelated to this work. R.-D. reported receiving grants from AstraZeneca, grants and non-financial support from Goldfinch Bio, being a scientific co-founder, consultant, and equity holder for Pensieve Health, and being a consultant for Variant Bio, all not related to this work. The remaining authors declare no competing interests.
