## [Peer Review File · Nature Communications]

Large-scale cross-ancestry genome-wide meta-analysis of serum urateREVIEWER COMMENTS

Reviewer #1 (Remarks to the Author):

In this study, Cho et al. meta-analyse GWAS results of serum urate (SU) from large cohorts (including the UK Biobank [UKB], the Chronic Kidney Disease Genetics Consortium [CKDGen] and the Korean Genome and Epidemiology Study [KoGES]). The key aims of the study are to (1) identify genetic variants associated with SU levels and the genes driving those associations, and (2) assess the role of SU in disease through PheWAS and MR analyses. The authors report new associations with SU, not only in ancestry-specific meta-analyses, but also in a cross-ancestry meta-analysis. In addition, the authors conduct a comprehensive set of downstream analyses that point to likely causal genes and assess the causal association between SU and three diseases with putative causal link (gout, hypertension, and heart failure).

Overall, the manuscript is well organised and well written. The analyses presented are relevant given the under-representation of non-EUR cohorts in GWAS SU genomic studies. There are, however, a few aspects that need further clarification, and I also provide some comments/suggestions that could help improve the manuscript.

1. I am particularly concerned with the GWAS conducted with UKB data.
 - a. These analyses used principal components, but no information is provided as to how these were computed. If the PCs used are the ones provided by the UKB, they may not be adequate, as they are based on the full set of individuals and capture mainly the large genetic differences between broad ancestry groups. They are not adequate to account for population stratification in Europeans for example.
 - b. The cross-ancestry meta-analysis includes a GWAS in non-European individuals from the UKB. This group is likely to include individuals that are genetically more distant from each other than from individuals of European ancestry, which were analysed in separate GWAS (eg, individuals of east Asian ancestry are more distant from African-ancestry individuals than European-ancestry individuals). Population stratification may be a problem in this GWAS.
 - c. Is it relevant to consider medications taken? In the introduction it is stated that urate-lowering therapeutics are associated with increased risk of cardiovascular death. This should be considered and/or discussed in analyses such as the survival test in UKB individuals.
2. In line 403 it is not clear if the 457,690 individuals used in the cross-ancestry meta-analysis were UKB participants. If so, why were some individuals with SU excluded? Please review the sentence.
3. Line 405, summary statistics of 110,739 individuals from KoGES were used in this study. Were the 22,607 individuals used for disease risk prediction not part of this discovery set?
4. Line 555, "Tenfold leave-one-out PRS (LOO PRS) was performed for each ancestry-specific genotype". This sentence makes it seem that PRS were calculated for each ancestry, but PRS was computed for exclusively for European individuals of the UKB. Please review.
5. Line 581, please review sentence.
6. Line 634, filtered out or selected?
7. 642-644, please review sentence.
8. Line 349, the association between SU and heart failure could be further discussed here – what are the conflicting interpretations?
9. Figure 2, legend does not describe panels b–d.
10. Please define acronyms used in Supplementary materials (eg SUA in ST19).
11. Please review references in the text (eg. line 345 not formatted).

Reviewer #2 (Remarks to the Author):

Cho et al. performed a transethnic meta-analysis of serum urate (SU) involving >1 million individuals

(677K Europeans, 220K East Asians, and 132K other ethnic groups). They identified 351 loci, including 17 previously unreported loci. The authors undertook a series of downstream bioinformatic analyses, including PRS PheWAS, in which a potential causal relationship with SU was observed for gout, heart failure, and hypertension. It is of note that an increased sample size has allowed for the discovery of new susceptibility loci for SU. Still, there are several points that the authors should take into consideration.

Major points:

1. A cross-ancestry GWAS claims all 17 previously unreported loci, presumably due to increased statistical power. However, it is unclear whether effect sizes at these loci are equivalent between European and East Asian meta-analyses. Please ensure this and put additional information in ST6, i.e., effect sizes and P-values separately by ethnic groups.
2. Apart from identifying 17 previously unreported loci, the novelty of the current meta-analysis is unclear to the reviewer. That is, 334 susceptibility loci (95% of 351 loci claimed in this study) have already been reported for previous SU GWASs, such as in CKDGen, UKBB, and KoGES samples. New samples appear to be added to GWAS meta-analysis in UK Biobank and KoGES. Please clarify in the text the number of newly added samples, if any. Although several downstream bioinformatic analyses are performed on the 351 loci, which parts are new and reanalyzed compared to previous SU GWASs must be clarified across the main text.
3. For a transethnic meta-analysis, it seems interesting to see whether ethnic groups other than Europeans and East Asians, i.e., South Asians in UKBB, show genetic associations similarly.
4. While the authors point out "the need for additional genomic studies conducted on non-European populations" in the Introduction, the corresponding detailed analysis and data interpretation seem insufficient to deepen our understanding of ethnic similarities and differences in SU genetics. Considering the relatively high genetic correlation of SU between European and East Asian ancestries, a cross-ancestry GWAS seems sufficient to identify SU-associated loci and related in silico analyses, with a few exceptions. Throughout the main text, results from three GWAS, cross-ancestry, European-only, and East Asian-only GWAS, are listed side-by-side without mention of biological significance. Accordingly, there is concern that this would distract the reader's attention.
5. Since an extensive PheWAS is currently feasible in UKBB alone, this part of the analysis is restricted to European populations, where the arguments about transethnic comparison do not fit.
6. This study's novelty and appealing points must be more sufficiently demonstrated in the Discussion. Instead, current descriptions of ULTs and their candidate genes are fragmented with no apparent context. Please consider re-writing them.

Minor Points:

1. Figure legends are missing for Fig.2b to Fig.2d.

Responses from the Authors to Review Comments:

We thank the editor and reviewers for their constructive comments and suggestions. We have performed extensive revisions to address them. Consequently, the figures and tables have been reordered due to the inclusion of additional supplementary materials.

REVIEWER COMMENTS:

Reviewer #1:

Remarks to the Author:

In this study, Cho et al. meta-analyse GWAS results of serum urate (SU) from large cohorts (including the UK Biobank [UKB], the Chronic Kidney Disease Genetics Consortium [CKDGen] and the Korean Genome and Epidemiology Study [KoGES]). The key aims of the study are to (1) identify genetic variants associated with SU levels and the genes driving those associations, and (2) assess the role of SU in disease through PheWAS and MR analyses. The authors report new associations with SU, not only in ancestry-specific meta-analyses, but also in a cross-ancestry meta-analysis. In addition, the authors conduct a comprehensive set of downstream analyses that point to likely causal genes and assess the causal association between SU and three diseases with putative causal link (gout, hypertension, and heart failure).

Overall, the manuscript is well organised and well written. The analyses presented are relevant given the under-representation of non-EUR cohorts in GWAS SU genomic studies. There are, however, a few aspects that need further clarification, and I also provide some comments/suggestions that could help improve the manuscript.

1. I am particularly concerned with the GWAS conducted with UKB data.

a. These analyses used principal components, but no information is provided as to how these were computed. If the PCs used are the ones provided by the UKB, they may not be adequate, as they are based on the full set of individuals and capture mainly the large genetic differences between broad ancestry groups. They are not adequate to account for population stratification in Europeans for example.

Response: We appreciate the reviewer's positive assessment of our work and their valuable comments. In our study, the PC values provided by the UK Biobank (UKBB) were used to perform the GWAS in European individuals (data-field 22009). To address the reviewer's concern, we performed an additional PCA using HapMap phase 3 variants on the unrelated European individuals (N= 276,250) from the UKBB who self-identified as 'White British' (data-field 21000) and have very similar genetic ancestry based on a PCA of the genotypes (data-field 22006). We then calculated the PCs of all UKBB European individuals (N = 408,188) using PC loadings from the PCA. GWAS results adjusted for the newly calculated PCs were highly consistent with the original GWAS results adjusted for the provided PCs (**Supplementary Fig. 2**).

The results are as follows:

Supplementary Fig. 2

[Added to the Results, page 5, lines 102–107]

To ascertain the robustness of the PC values provided by the UKBB in accounting for

population stratification, we also performed GWAS using PC values derived from European and non-European populations, respectively (**Methods**). GWAS results adjusted for the newly calculated PCs were highly consistent with the original GWAS results adjusted for the provided PCs (**Supplementary Figs. 2 and 3**).

[Added to the Methods, page 21, lines 488–497]

The first four principal components (PCs) of genetic ancestry, which were calculated based on the entire UKBB population provided by the UKBB (data field 22009), were used as covariates. To examine the robustness of the PCs in accounting for population stratification, we additionally performed a PCA using HapMap phase 3 variants on the unrelated European individuals ($N = 276,250$) from the UKBB who self-identified as 'White British' (data-field 21000) and have very similar genetic ancestry based on a PCA of the genotypes (data-field 22006). We then calculated the PCs of all UKBB European individuals ($N = 408,188$) using PC loadings from the PCA. The Spearman's correlation coefficient was used to compare the beta coefficients from the GWAS of the two sets of PCs (**Supplementary Fig. 2**).

b. The cross-ancestry meta-analysis includes a GWAS in non-European individuals from the UKB. This group is likely to include individuals that are genetically more distant from each other than from individuals of European ancestry, which were analysed in separate GWAS (eg, individuals of east Asian ancestry are more distant from African-ancestry individuals than European-ancestry individuals). Population stratification may be a problem in this GWAS.

Response: We appreciate the reviewer's valuable comment regarding the population stratification of non-European individuals. To address the reviewer's concern, we defined seven genetically distinct groups for individuals categorized as non-White British" ancestry based on the PC values provided by the UKBB, as delineated by Privé et al. (AJHG, 2022). We performed GWAS of serum urate (SU) in each of the seven genetically distinct groups separately and analyzed the results. The meta-analysis of the seven non-European GWAS ($n = 28,320$) showed highly consistent effect sizes of the analyzed variants, but slightly less

significant associations in comparison with the combined non-European GWAS adjusted for PCs provided by the UKBB (n = 72,170). Based on this observation, we maintained the main and downstream analyses by utilizing 72,170 non-European individuals to enhance the statistical power.

The results are as follows:

Supplementary Fig. 3

[Added to the Results, page 5, lines 102–107]

To ascertain the robustness of the PC values provided by the UKBB in accounting for population stratification, we also performed GWAS using PC values derived from European and non-European populations, respectively (**Methods**). GWAS results adjusted for the newly calculated PCs were highly consistent with the original GWAS results adjusted for the provided PCs (**Supplementary Figs. 2 and 3**).

[Added to the Methods, page 21, lines 497–506]

To examine population stratification in the GWAS of UKBB non-European individuals, we defined seven genetically distinct groups for individuals categorized as having non-White British" ancestry based on the PC values provided by UKBB, as delineated by Privé et al.⁸⁵. We performed a GWAS of SU in each of the seven genetically distinct groups separately and analyzed the results. The meta-analysis of seven non-European GWAS ($N = 28,320$) showed highly consistent effect sizes of the analyzed variants, but slightly less significant associations in comparison with the combined non-European GWAS adjusted for PCs provided by the UKBB ($N = 72,170$) (**Supplementary Fig. 3**). Based on this observation, the analyses in this study were conducted using 72,170 non-European individuals from the UKBB to enhance the statistical power.

c. Is it relevant to consider medications taken? In the introduction it is stated that urate-lowering therapeutics are associated with increased risk of cardiovascular death. This should be considered and/or discussed in analyses such as the survival test in UKB individuals.

Response: Thank you for this valuable suggestion. Following the reviewer's advice, we excluded UKBB individuals who took ULT-related medications at enrollment (data field 20003; allopurinol and probenecid) from the survival analyses. We replaced the previous results with these new results in the revised manuscript, although no significant differences were observed in the survival analysis results.

[Added to the Results, page 10, lines 219–223]

To investigate the association between the polygenic risk for SU and the risk of gout, heart failure, and hypertension, we performed survival analyses using cross-ancestry and the European PRS on 380,213 participants with gout, 331,432 participants with heart failure, and 357,453 participants with hypertension who did not take ULT-related medications at enrollment.

[Added to the Results, page 10, lines 227–234]

Cross-ancestry and European ancestry SU PRS were significantly associated with gout risk (cross-ancestry, hazard ratio [HR] = 1.63; 95% confidence interval [95% CI] = 1.592–1.67; $P < 2.00 \times 10^{-16}$; European, HR = 1.641; 95% CI = 1.603–1.681; $P < 2.00 \times 10^{-16}$). Both SU PRS were also significantly associated with risk of heart failure (cross-ancestry, HR = 1.056; 95% CI = 1.036–1.078; $P < 2.00 \times 10^{-16}$; European, HR = 1.061; 95% CI = 1.04–1.082; $P < 2.00 \times 10^{-16}$) and hypertension (cross-ancestry, HR = 1.065; 95% CI = 1.056–1.073; $P < 2.00 \times 10^{-16}$; European, HR = 1.065; 95% CI = 1.057–1.074; $P < 2.00 \times 10^{-16}$).

[Added to the Results, page 10, lines 236–239]

For example, participants in the very high (99th percentile) SU cross-ancestry PRS group showed 7.00-, 1.37-, and 1.34-times higher risk of gout, heart failure, and hypertension, respectively, than participants in the low (0–19th percentile) SU cross-ancestry PRS group.

[Added to the Discussion, page 17, lines 387–390]

This study found that individuals with very high (99th percentile) SU cross-ancestry PRS had 7.00-, 1.37-, and 1.34-fold higher risks of gout, heart failure, and hypertension, respectively, than those with low (0–19th percentile) SU cross-ancestry PRS.

[Added to the Methods, pages 29–30, lines 706–707]

We excluded UKBB individuals who took ULT-related medications at enrollment (data field 20003; allopurinol and probenecid) from the survival analyses.

2. In line 403 it is not clear if the 457,690 individuals used in the cross-ancestry meta-analysis were UKB participants. If so, why were some individuals with SU excluded? Please review the sentence.

Response: Thank you for the comment. Our initial description of the summary statistics of the CKDGen was unclear. The 457,690 individuals used in the cross-ancestry meta-analysis were provided by the CKDGen and were not included in the UKBB. We have clarified these statements in the revised manuscript to avoid confusion. The number of participants and data used for each analysis are shown in **Fig. 1**.

[Added to the Methods, page 20, lines 468–470]

Summary statistics for the cross-ancestry GWAS (457,690 individuals) and the European GWAS (288,649 individuals) were provided by the CKDGen, and the individuals in both datasets were not included in the UKBB.

3. Line 405, summary statistics of 110,739 individuals from KoGES were used in this study. Were the 22,607 individuals used for disease risk prediction not part of this discovery set?

Response: Thank you for the comment. A total of 110,739 individuals from the KoGES were independent of the 22,607 individuals used in the disease risk prediction analysis. We clarified

this in the revised manuscript to avoid confusion.

[Added to the Methods, page 20, lines 473–474]

Additionally, 22,607 individuals, independent of the 110,739 individuals involved in the discovery analysis, were included in the replicates.

[Added to the Methods, page 28, lines 671–672]

This dataset was independent from the 110,739 KoGES individuals used in the discovery analysis.

4. Line 555, “Tenfold leave-one-out PRS (LOO PRS) was performed for each ancestry-specific genotype”. This sentence makes it seem that PRS were calculated for each ancestry, but PRS was computed for exclusively for European individuals of the UKB. Please review.

Response: We appreciate the reviewer’s comment regarding this confusion. This sentence was incorrectly stated in our previous manuscript since we performed an analysis on European individuals of the UKBB. However, we have retained this in the revised manuscript because we additionally performed PRS PheWAS in the Korean population using LOO PRS for KoGES individuals, as requested by Reviewer #2.

[Added to the Methods, page 28, lines 660–666]

Similarly, we performed LOO PRS for East Asian ancestry. The KoGES data of 72,299 of the 110,739 unrelated individuals, whose individual-level genotype data were available in this study, were randomly divided into ten groups. Association analysis was performed on nine datasets using PLINK, and PRS was calculated for the remaining dataset. As a result, the cross-

ancestry and East Asian PRS for 72,299 individuals from the KoGES were obtained. The Pearson correlation coefficients between the PRS and SU residuals were 0.272 and 0.267 for the cross-ancestry and East Asian PRS, respectively.

5. Line 581, please review sentence.

Response: We appreciate the reviewer's comments regarding this confusion. This sentence explains our methodology for estimating odds ratio across PRS decile groups compared to the reference group (the first decile group). We have clarified this sentence in the revised manuscript to avoid confusion.

[Added to the Methods, pages 28–29, lines 681–682]

We partitioned the individuals into PRS deciles and estimated the OR between the first decile and each of the other decile groups.

6. Line 634, filtered out or selected?

Response: Thank you for this comment. Genes identified from the DEPICT, TWAS, and colocalization analyses were considered for SMR analysis; therefore, 'selected' was correct. We have modified the text accordingly.

[Added to the Methods, page 31, lines 740–742]

We gathered SU-associated genes from DEPICT, TWAS, and colocalization analysis and then

selected the same genes that had expression data in the NEPTUNE kidney tissue.

7. 642-644, please review sentence.

Response: Thank you for this comment. We performed an MR analysis to examine the potential consistent causality between SU and the traits of interest. Consequently, we filtered out genes whose effect (beta) directions on SU and the trait differed and focused on genes that consistently influenced both SU levels and the trait. Following the reviewer's advice, we revised the sentence accordingly.

[Added to the Methods, pages 31, lines 750–752]

Based on the MR results, which indicated that high SU levels were associated with an increased risk of gout, heart failure, and hypertension, we excluded genes that showed the opposite effects on SU and disease.

8. Line 349, the association between SU and heart failure could be further discussed here – what are the conflicting interpretations?

Response: We thank the reviewer for their valuable comments on the statements that require further discussion. Although MR analysis supported an association between SU and heart failure in the current study, conflicting interpretations of this relationship exist in the scientific community. The relationship between SU and gout is well established and serves as a clear example of a direct causal relationship, whereas the relationship between SU and hypertension and heart failure remains controversial. Stewart et al. highlighted the difficulty in inferring the

direct causality of hyperuricemia on hypertension and suggested that a larger-scale randomized controlled trial is required to further elucidate this relationship. Regarding heart failure, the study by Krishnan et al. was pivotal in identifying SU as a potential risk factor, which had previously been unrecognized. However, large-scale MR studies have failed to identify a causal relationship between SU and heart failure. MR analysis in the current study has provided additional evidence supporting that SU may play a role in the development of heart failure. We revised the manuscript to describe these findings more clearly.

[Added to the Discussion, pages 17–18, lines 395–414]

SU has been extensively studied, with most studies confirming its direct causative role in gout. However, its role in other diseases remains controversial. Stewart et al. highlighted the difficulty in inferring a direct causal relationship between hyperuricemia and hypertension and suggested that large-scale randomized trials are required to further elucidate this relationship⁵⁵. A study by Krishnan et al. was pivotal in identifying SU as a potential risk factor for heart failure, which had previously been unrecognized⁵⁶. However, large-scale MR studies have failed to identify a causal relationship among SU, blood pressure, and heart failure^{57,58}. In addition, umbrella reviews of SU have failed to find convincing evidence regarding the clear role of SU in diseases other than gout and nephrolithiasis¹. In contrast to these negative findings, a recent MR study showed that genetically determined SU levels were significantly associated with heart failure (OR = 1.07, 95% CI = 1.03–1.10; $P = 8.6 \times 10^{-5}$)⁵⁹. The current study replicated this result using more instrumental variables (OR = 1.10; 95% CI = 1.05–1.16; $P = 1.78 \times 10^{-4}$). Although associations between SU and hypertension have been reported^{60,61}, a causal genetic association has not yet been established in MR research^{57,62}. Our study demonstrated a potential causal association between SU and hypertension without genetic pleiotropy (OR = 1.20; 95% CI = 1.11–1.30; $P = 2.80 \times 10^{-6}$); this is in contrast to a previous report about the existence of genetic pleiotropy⁶³. This result provided evidence that a direct causal relationship may exist without the pleiotropic effect of SU on hypertension, which is consistent with previous studies that suggested a causal relationship between SU and BP and gout with hypertension, respectively^{64,65}.

9. Figure 2, legend does not describe panels b–d.

Response: Thank you for your comment. We have added legends for these panels to **Fig. 2**.

[Added to the Fig. 2]

Fig. 2 Mirrored Manhattan plots of GWAS and TWAS in the cross-ancestry meta-analysis and comparison of the variant effect sizes for each ancestry-specific meta-analysis.

(b, c, d) Comparison of effect sizes of the lead variants between the European and East Asian ancestries. b, Cross-ancestry meta-analysis lead variants. c, European ancestry meta-analysis lead variants. d, East Asian ancestry meta-analysis lead variants.

10. Please define acronyms used in Supplementary materials (eg SUA in ST19).

Response: Thank you for your comment. We have added the correct abbreviations throughout the manuscript, Figures, and Tables.

11. Please review references in the text (eg. line 345 not formatted).

Response: Thank you for your comment. We have reviewed all the references and revised any formatting errors.

We sincerely appreciate the valuable time and effort you have dedicated to reviewing our manuscript.

Reviewer #2:

Remarks to the Author:

Cho et al. performed a transethnic meta-analysis of serum urate (SU) involving >1 million individuals (677K Europeans, 220K East Asians, and 132K other ethnic groups). They identified 351 loci, including 17 previously unreported loci. The authors undertook a series of downstream bioinformatic analyses, including PRS PheWAS, in which a potential causal relationship with SU was observed for gout, heart failure, and hypertension. It is of note that an increased sample size has allowed for the discovery of new susceptibility loci for SU. Still, there are several points that the authors should take into consideration.

Major points:

1. A cross-ancestry GWAS claims all 17 previously unreported loci, presumably due to increased statistical power. However, it is unclear whether effect sizes at these loci are equivalent between European and East Asian meta-analyses. Please ensure this and put additional information in ST6, i.e., effect sizes and P-values separately by ethnic groups.

Response: We sincerely appreciate the reviewers' positive remarks on the value of our study and their constructive comments. Following the reviewer's suggestion, we have added summary statistics, including the effect size, standard error, allele frequency, and P-values by ethnic group in **Supplementary Table 6**. We observed that the lead variants from the cross-ancestry meta-analysis, including 17 previously unreported variants, had similar effect sizes in the European and East Asian ancestries (see Fig. 2b). We have also included additional descriptions in the Discussion section to emphasize this finding.

[Added to the Discussion, page 13, lines 305–306]

We observed similar effect sizes for these loci between the European and East Asian

populations.

2. Apart from identifying 17 previously unreported loci, the novelty of the current meta-analysis is unclear to the reviewer. That is, 334 susceptibility loci (95% of 351 loci claimed in this study) have already been reported for previous SU GWASs, such as in CKDGen, UKBB, and KoGES samples. New samples appear to be added to GWAS meta-analysis in UK Biobank and KoGES. Please clarify in the text the number of newly added samples, if any. Although several downstream bioinformatic analyses are performed on the 351 loci, which parts are new and reanalyzed compared to previous SU GWASs must be clarified across the main text.

Response: We appreciate the valuable comment from the reviewer. We acknowledge that the Discussion section of our manuscript does not describe novel points or the aims of our study. This study aimed to examine the shared genetic architecture of SU by performing and comparing ancestry-specific meta-analyses and identifying novel loci by incorporating additional cohorts into existing cross-ancestry SU GWAS. Although most of the identified SU loci have been reported previously, 17 previously unreported loci were identified using highly stringent criteria. These loci were at least 2Mb away from previously reported loci. In addition, we investigated the causal relationships among genes, diseases, and SU. In response to the reviewer's recommendations, we have revised the manuscript to include more detailed explanations of our novel findings and study aims.

[Added to the Discussion, pages 13–14, lines 300–322]

We conducted a large-scale cross-ancestry meta-analysis of 460,894 individuals from the UKBB and 110,739 individuals from KoGES. This extended study included 1,029,323 individuals, which is approximately double the sample size of previous cross-ancestry studies. In this study, we identified 351 significant SU-associated genetic loci, including 17 previously unreported loci^{13,32,33}, which were more than 2Mb away from the previously reported loci. We observed similar effect sizes for these loci between the European and East Asian populations.

These SU-associated loci are enriched in the SU-related tissues, including the urinary tract and kidney in the urogenital system. Our GWAS meta-analysis provided additional insights that have not been thoroughly examined in previous SU studies. 1) The SU-associated loci showed similar effect sizes, high genetic correlation, and shared genetic architecture across ancestries, which was in line with the findings on the traits from other studies^{34,35}. The effect sizes of the lead variants were positively correlated ($\rho = 0.426\text{--}0.77$) across the European, East Asian, and other ancestries. 2) In addition to the cross-ancestry analysis, we conducted downstream analyses based on GWAS results for each ancestry, which allowed us to identify ancestry-specific results in some analyses. 3) We identified 467 and 323 potential causal genes in the tubulointerstitial and glomerular kidney tissues, respectively, among 2,671 genes in the SU-associated GWAS loci, through a series of enrichment analysis, colocalization analysis, TWAS, and SMR. 4) In the PheWAS with PRS, the PRS of SU was significantly associated with gout, heart failure, and hypertension. We identified the significant potential causal effects of SU on these SU-associated diseases, including heart failure and hypertension, which was previously controversial. 5) We identified ten genes that showed potential causal associations of SU along with heart failure and hypertension and investigated their effects on the diseases through SU.

3. For a transethnic meta-analysis, it seems interesting to see whether ethnic groups other than Europeans and East Asians, i.e., South Asians in UKBB, show genetic associations similarly.

Response: Thank you for this valuable suggestion. Accordingly, we performed a GWAS of SU for each genetically distinct group of the UKBB non-European populations separately and compared these results with those from the cross-ancestry GWAS. We defined seven genetically distinct groups for individuals categorized as non-White British" ancestry based on the PC values provided by UKBB, as delineated by Privé et al.⁸⁵. We analyzed four groups with a minimum sample size of 3,000 individuals to ensure sufficient statistical power. We performed a GWAS of SU in each genetically distinct group separately. To assess the directional consistency of the genetic effects, we compared the beta coefficients of the lead variants from the cross-ancestry GWAS with those from each group in the non-European

population using Spearman’s correlation and Cohen’s kappa coefficients. We observed positive correlations overall ($\rho = 0.426–0.77$), despite the limited sample sizes. We suggest further studies in non-European populations with enriched sample sizes be conducted to achieve more conclusive outcomes.

The results are as follows:

Supplementary Fig. 4

[Added to the Results, page 6 lines 115–118]

We compared the effect size and the direction of the lead variants in the cross-ancestry meta-

analysis with those in each of the four genetically distinct groups (India, Italy, Nigeria, and Poland) and found significant positive correlations ($\rho = 0.426\text{--}0.77$, $\kappa = 0.376\text{--}0.626$, **Supplementary Fig. 4**).

[Added to the Discussion, page 14, lines 311–312]

The effect sizes of the lead variants were positively correlated ($\rho = 0.426\text{--}0.77$) across the European, East Asian, and other ancestries.

[Added to the Methods, pages 24–5, lines 575–582]

We compared the effect sizes of the lead variants in the cross-ancestry GWAS meta-analysis with those from the GWAS of each of the four genetically distinct groups in UKBB with a sample size of >3000 individuals. The GWAS was performed for each group using the same QC process that was used for the UKBB European GWAS. Among the lead variants in the cross-ancestry GWAS meta-analysis, 263, 323, 177, and 331 variants were found in the data from India, Italy, Nigeria, and Poland, respectively. Spearman’s correlation and Cohen’s kappa coefficients were used to compare the effect sizes and directional consistencies of the genetic effects, respectively.

4. While the authors point out “the need for additional genomic studies conducted on non-European populations” in the Introduction, the corresponding detailed analysis and data interpretation seem insufficient to deepen our understanding of ethnic similarities and differences in SU genetics. Considering the relatively high genetic correlation of SU between European and East Asian ancestries, a cross-ancestry GWAS seems sufficient to identify SU-associated loci and related in silico analyses, with a few exceptions. Throughout the main text, results from three GWAS, cross-ancestry, European-only, and East Asian-only GWAS, are listed side-by-side without mention of biological significance. Accordingly, there is concern that this would distract the reader's attention.

Response: We appreciate the reviewer's concern regarding the absence of biological comparisons between the ancestry and the related descriptions. As mentioned by the reviewer, a cross-ancestry GWAS increases the statistical power to identify SU-associated loci and colocalized genes. In addition to the cross-ancestry analysis, ancestry-specific findings were identified through analyses based on the GWAS results for each ancestry.

- (Results page 7, lines 152–153) Cardiovascular system-related tissues, such as heart valves, were significantly enriched in the European ancestry only ($P = 6.07 \times 10^{-4}$).
- (Results page 9, lines 197–199) While 178 genes were commonly significant in the three meta-analyses, 183 of 945, 83 of 780, and 75 of 334 significant genes were only significant in the cross-ancestry, European, and East Asian populations, respectively.
- (Discussion page 15, lines 348–350) East Asian ancestry analysis identified the association of SU with the MAPK signaling pathway, supported by previous findings that SU is associated with renal tissue growth through the MAPK pathway⁴³.

We recognize that the ancestry-specific findings listed above were present in the original manuscript but were not sufficiently highlighted and discussed. We have added to and further discussed the ancestry-specific results.

[Added to the Results, page 7, lines 157–159]

The fetal blood of the hematologic and immune system ($P = 9.31 \times 10^{-3}$) and the nasal mucosa of the respiratory system ($P = 3.65 \times 10^{-3}$) were significantly enriched in the East Asian ancestry only.

[Added to the Results, page 8, lines 179–180]

Of the colocalized genes, 27 and 9 were identified in the European and East Asian ancestry only, respectively.

[Added to the Discussion, pages 13–14, lines 31–314]

In addition to the cross-ancestry analysis, we conducted downstream analyses based on the GWAS results for each ancestry, which allowed some analyses to identify ancestry-specific results.

[Added to the Discussion, page 14, lines 323–324]

Colocalization analysis identified 173 potentially causal genes in the SU-associated loci, including 36 genes identified in the ancestry-specific analysis.

[Added to the Discussion, pages 14–15, lines 332–343]

The cross-ancestry meta-analysis results were enriched in most tissues, including the urinary tract of the urogenital system, kidney, and cartilage and exhibited the lowest P-values. In addition, we observed ancestry-specific enrichment in tissues such as the cardiovascular tissues in European ancestry and the fetal blood and the nasal mucosa tissues in East Asian ancestry. The enrichment in the cardiovascular tissues was consistent with the results of the MR analysis of the European ancestry, which identified a potential causal relationship between SU and both heart failure and hypertension. The East Asian meta-analysis results were enriched in the fetal blood of the hematologic and immune system ($P = 0.001$) and the nasal mucosa of the respiratory system ($P = 3.65 \times 10^{-3}$). Previous clinical studies have shown that SU is associated with fetal growth^{36,37}, and studies on the association between various air pollutants and the nasal cavity have revealed urate as an important first-line defense factor against reactive oxygen species^{38,39}.

[Added to the Discussion, pages 15–16, lines 359–361]

Only the East Asian ancestry analysis revealed an association between SU and the MAPK signaling pathway, supported by previous findings that SU is associated with renal tissue growth through the MAPK pathway⁴⁷.

[Added to the Discussion, pages 15–16, lines 358–368]

The ancestry-specific findings in this study have two possible explanations. It is possible that ancestry-specific genetic loci affect SU-related biological pathways in certain ancestral populations only, or that the identification of such unique loci and the subsequent findings based on them may also be due to the differences in statistical power in each ancestry. For example, despite shared biological mechanisms across ancestries, some genetic loci can be identified as ancestry-specific loci that are unidentifiable in other ancestries or cross-ancestry meta-analysis, owing to several factors, such as different allele frequencies, LD structure, and environmental factors. Therefore, the interpretation of ancestry-specific findings requires caution, and comparisons are warranted for larger datasets across ancestries. Nevertheless, genetic studies of diverse ancestries and the findings from each ancestry may provide new and valuable insights into the biological background of SU.

5. Since an extensive PheWAS is currently feasible in UKBB alone, this part of the analysis is restricted to European populations, where the arguments about transethnic comparison do not fit.

Response: We appreciate the reviewer’s valuable comment. For a transethnic comparison, we conducted LOO PRS analyses and PheWAS with PRS in KoGES for 72,299 of the 110,739 unrelated individuals whose individual-level genotype data were available. PheWAS was performed using a Firth’s bias-reduced logistic regression model for 37 self-reported diseases in the KoGES. Consistent with the results from the European population, gout and hypertension were significantly associated with the SU PRS in the PheWAS of the Korean population.

The results are as follows:

Supplementary Table 18

Supplementary Table 18. Significant results of the PRS phenome-wide association study using cross-ancestry PRS and East Asian ancestry PRS in the Korean population (Bonferroni correction)

Phenotype	OR	P-value*	R ² **	PRS source
Hypertension	1.055	1.31×10^{-6}	0.00058	East Asian PRS
Gout	1.442	3.95×10^{-11}	0.009748	East Asian PRS
Hypertension	1.108	5.25×10^{-20}	0.002069	Cross ancestry PRS
Gout	1.550	2.15×10^{-18}	0.015805	Cross ancestry PRS

* The P-value of PRS corrected for age and sex.

** The R² is the Nagelkerke pseudo R² of PRS.

[Added to the Result, pages 9, lines 214–218]

In addition, we conducted the PheWAS with the cross-ancestry SU PRS and East Asian SU PRS on the Korean participants to investigate the similarities and differences with the European results (Methods and Supplementary Fig. 6b). Among the 37 self-reported diseases, gout and hypertension were significantly associated with the SU PRS (Supplementary Table 18).

[Added to the Methods, page 28, lines 658–664]

Similarly, we performed LOO PRS for the East Asian ancestry. The KoGES data of 72,299 of the 110,739 unrelated individuals, whose individual-level genotype data were available in this study, were randomly divided into ten groups. Association analysis was performed on nine datasets using PLINK, and the PRS was calculated for the remaining dataset. As a result, cross-ancestry and East Asian PRS for 72,299 individuals from the KoGES were obtained. The Pearson correlation coefficients between the PRS and SU residuals were 0.272 and 0.267 for the cross-ancestry and East Asian PRS, respectively.

[Added to the Methods, page 29, lines 687–688]

Phenome-wide association study in European and East Asian populations using cross-ancestry and ancestry-specific PRS

[Added to the Methods, page 29, lines 695–700]

We conducted the PheWAS with PRS in an East Asian population using a similar approach. Cross-ancestry and East Asian PRS for KoGES individuals obtained from the LOO PRS were adjusted for age, sex, and the first four PCs. PheWAS was performed using Firth’s bias-reduced logistic regression model for 37 self-reported diseases in the KoGES. We considered results with P-values less than the Bonferroni correction threshold ($P_{\text{bon}} = 0.05/37$) to be significant.

[Added to the Discussion, pages 16, lines 380–382]

Consistent with the results from the European population, gout and hypertension were significantly associated with the SU PRS in the PheWAS of the Korean population.

6. This study's novelty and appealing points must be more sufficiently demonstrated in the Discussion. Instead, current descriptions of ULTs and their candidate genes are fragmented with no apparent context. Please consider rewriting them.

Response: We appreciate the reviewer’s valuable comments. We recognize the importance of further explaining why we chose this particular research design, detailing the specific analyses we performed and explaining the implications of their results. Accordingly, we have revised and restructured the relevant sections to improve clarity and context.

[Added to the Discussion, page 18, lines 415–433]

For gout, heart failure, and hypertension, which were potentially causally associated with SU, SMR analysis was performed to infer the association between these traits and the expression of 2,671 SU-related genes selected from the enrichment analysis, colocalization analysis, and TWAS. The associations between these genes and traits were investigated to identify new treatment targets for these three diseases. Typically, the ULT methods employed to reduce SU

levels involve the use of drugs with specific mechanisms of action (MOA), such as xanthine oxidase inhibitors (XOIs), uricosuric agents, and uricase. The 2020 ACR guidelines recommends XOIs as the first-line treatment for patients with gout⁵. However, only two drugs (allopurinol and febuxostat) are widely used as XOIs. Several large randomized clinical trials have shown that allopurinol is ineffective in the treatment of hypertension, CKD, and ischemic heart disease^{66–70}. This indicates that ULT drugs are almost exclusively effective in the treatment of gout. A review of all currently available ULTs highlights the need for new ULTs with multiple mechanisms⁶. Our study identified candidate genes for new ULT methods (four for gout, one for heart failure, and five for hypertension) that may have a direct causal effect on SU and an indirect effect on the three diseases via SU. For these genes, the proportion of the mediation effect of SU on gout, heart failure, and hypertension was examined and found to be smaller than that of the direct effect. These genes may have multiple mechanisms in these three diseases, including direct and indirect effects via SU. Further research is required to elucidate the functions of these genes.

[Added to the Discussion, pages 18–19, lines 434–436]

Although a direct association between these genes and SU has not been reported, this potential relationship is supported by other biological experimental studies. We investigated the MOA of the ten ULT candidate genes identified in our study.

[Added to the Discussion, page 19, lines 448–450]

Although a direct functional relationship between these genes and SU remains unclear, further studies are required to identify the precise biological mechanisms underlying these potential target genes.

[Added to the Discussion, page 19, lines 453–455]

This approach highlighted the potential of repositioned drugs targeting SU for the treatment of other diseases. In addition, we identified potential causal relationships between SU, target

genes, and various diseases.

Minor Points:

1. Figure legends are missing for Fig.2b to Fig.2d.

Response: Thank you for your comment. We have added legends to the panels in **Fig. 2**.

[Added to the Fig. 2]

Fig. 2 Mirrored Manhattan plots of GWAS and TWAS in the cross-ancestry meta-analysis and comparison of the variant effect sizes for each ancestry-specific meta-analysis

(b, c, d) Comparison of the effect sizes of the lead variants between the European and East Asian ancestries. b Cross-ancestry meta-analysis of lead variants. c European ancestry meta-analysis lead variants. d East Asian ancestry meta-analysis of lead variants.

We sincerely appreciate the valuable time and effort you have dedicated to reviewing our manuscript.

REVIEWERS' COMMENTS

Reviewer #1 (Remarks to the Author):

First, thank you to the authors for the additional analyses included in the manuscript. I appreciate the explanation of how analyses were performed and the added results, which provide more confidence in the findings presented in the main manuscript. I have a only couple of follow-up comments for clarification.

- For the analyses of non-European individuals, the authors defined seven genetically-predicted ancestry groups and conducted separate GWAS for each group. It is unclear, however, whether these analyses were controlled for within-ancestry PCs. Please provide more detail.
- Please clarify method used to compute PCs in sensitivity analyses.

Reviewer #2 (Remarks to the Author):

The reviewers have addressed most of the critiques and revised the manuscript accordingly.

I have only one minor comment. The authors used Cohen's kappa coefficients, in addition to Spearman's correlation, in Fig. S4. Since Cohen's kappa coefficients are used for categorical items, Spearman's correlation is sufficient in this case.

Responses from the Authors to Review Comments:

We thank the editor and reviewers for their constructive comments and suggestions. We have added sentences to the manuscript to address the remaining comments with appropriate responses.

REVIEWER COMMENTS:

Reviewer #1:

Remarks to the Author:

First, thank you to the authors for the additional analyses included in the manuscript. I appreciate the explanation of how analyses were performed and the added results, which provide more confidence in the findings presented in the main manuscript. I have a only couple of follow-up comments for clarification.

- For the analyses of non-European individuals, the authors defined seven genetically-predicted ancestry groups and conducted separate GWAS for each group. It is unclear, however, whether these analyses were controlled for within-ancestry PCs. Please provide more detail.
- Please clarify method used to compute PCs in sensitivity analyses.

Response: Thank you for the reviewer's positive remark on the revised manuscript and valuable comment. We calculated PCs for each non-European group and used them as covariates in the GWAS. Detailed information on this has been added to the Methods section.

[Added to the Methods, page 21, lines 501–504]

Association analysis was performed using linear mixed model analysis with SAIGE v1.1.3 as the residual value of SU (mg/dl) \approx AGE + SEX. The first four principal components (PCs) of

genetic ancestry, calculated for each ancestry group using PLINK, were used as covariates.

We are very grateful for your valuable time and effort in reviewing our manuscript.

Reviewer #2:

Remarks to the Author:

The reviewers have addressed most of the critiques and revised the manuscript accordingly.

I have only one minor comment. The authors used Cohen's kappa coefficients, in addition to Spearman's correlation, in Fig. S4. Since Cohen's kappa coefficients are used for categorical items, Spearman's correlation is sufficient in this case.

Response: We appreciate the reviewer's positive evaluation on our work and valuable comment. We aimed to assess not only the linear relationship but also the concordance of effect size direction across study populations. As mentioned by the reviewer and utilized in prior studies (Nat Hum Behav, 2022 and Sci Rep, 2020), Cohen's kappa coefficients were used to assess the concordance of effect size direction (categorical variable).

We sincerely appreciate the valuable time and effort you have dedicated to reviewing our manuscript.

References

Kim, S. *et al.* Shared genetic architectures of subjective well-being in East Asian and European ancestry populations. *Nat. Hum. Behav.* **6**, 1014-1026 (2022).

Cho, S. K. *et al.* Polygenic analysis of the effect of common and low-frequency genetic variants on serum uric acid levels in Korean individuals. *Sci. Rep.* **10**, 9179 (2020).